# Crystal structure of an archaeal CorB magnesium transporter

Yu Seby Chen [1], Guennadi Kozlov [1], Brandon E. Moeller[2], Ahmed Rohaim[3,4], Rayan Fakih[1], Benoît Roux [3], John E. Burke [2,5] & Kalle Gehring [1✉]

CNNM/CorB proteins are a broadly conserved family of integral membrane proteins with close to 90,000 protein sequences known. They are associated with $Mg^{2+}$ transport but it is not known if they mediate transport themselves or regulate other transporters. Here, we determine the crystal structure of an archaeal CorB protein in two conformations (apo and $Mg^{2+}$-ATP bound). The transmembrane DUF21 domain exists in an inward-facing conformation with a $Mg^{2+}$ ion coordinated by a conserved π-helix. In the absence of $Mg^{2+}$-ATP, the CBS-pair domain adopts an elongated dimeric configuration with previously unobserved domain-domain contacts. Hydrogen-deuterium exchange mass spectrometry, analytical ultracentrifugation, and molecular dynamics experiments support a role of the structural rearrangements in mediating $Mg^{2+}$-ATP sensing. Lastly, we use an in vitro, liposome-based assay to demonstrate direct $Mg^{2+}$ transport by CorB proteins. These structural and functional insights provide a framework for understanding function of CNNMs in $Mg^{2+}$ transport and associated diseases.

[1] Department of Biochemistry & Centre de Recherche en Biologie Structurale, McGill University, Montréal, QC, Canada. [2] Department of Biochemistry and Microbiology, University of Victoria, Victoria, BC, Canada. [3] Department of Biochemistry and Molecular Biology, University of Chicago, Gordon Center for Integrative Science, Chicago, IL, USA. [4] Department of Biophysics, Faculty of Science, Cairo University, Giza, Egypt. [5] Department of Biochemistry and Molecular Biology, The University of British Columbia, Vancouver, BC, Canada. ✉email: kalle.gehring@mcgill.ca

Magnesium ($Mg^{2+}$), the most abundant divalent cation inside cells, is essential for a wide variety of biochemical processes, such as energy metabolism, maintenance of genomic stability, protein synthesis, and over 600 enzymatic reactions[1]. CNNMs (CBS-pair domain divalent cation transport mediators) are a conserved family of integral membrane proteins implicated in $Mg^{2+}$ homeostasis and divalent cation transport[2]. CNNM2 and CNNM4 are found abundantly in the basolateral membrane of kidney and colon epithelial cells, where renal/intestinal (re)absorption of $Mg^{2+}$ occurs[3,4]. In humans, mutations in CNNM proteins are linked to two genetic diseases: CNNM2 mutations cause hypomagnesemia[3,5] while mutations in CNNM4 are associated with Jalili syndrome[6]. CNNM2-knockout mice are embryonic lethal[7], while loss of CNNM4 leads to male infertility[8] and susceptibility to cancer[9,10]. CNNMs are additionally implicated in hypertension[7], non-alcoholic steatohepatitis[11], and schizophrenia[12].

CNNMs are also known as ancient conserved domain proteins (ACDPs) because their core domains are evolutionarily conserved in essentially all organisms, from animals, plants, fungi, and archaea to bacteria (Fig. 1a)[13]. The bacterial CorB protein was first identified in a screen for cobalt-resistant mutants in Salmonella typhimurium[14]. CorB was proposed to mediate $Mg^{2+}$ efflux together with CorC and CorD, in which CorC is a soluble protein that shares high sequence similarity to the cytosolic domains of CorB. Subsequently, many other CorB orthologs have been implicated in $Mg^{2+}$ transport. For example, the Staphylococcus aureus ortholog, MpfA, is thought to function as a $Mg^{2+}$ exporter as deletion mutants are unable to grow in the presence of high concentrations of magnesium[15,16]. Disruption of the homologous gene (yhdP) in Bacillus subtilis leads to increased cellular $Mg^{2+}$ content, again supporting a role in $Mg^{2+}$ efflux[17].

Despite the clear association with $Mg^{2+}$ transport, the field has not yet reached a consensus on the molecular mechanism of CNNM/CorB proteins[18,19]. Some papers report direct efflux of $Mg^{2+}$ through $Na^+$-coupled exchange[4], others observed direct $Mg^{2+}$ influx[20,21], while still others propose an indirect mechanism through regulation of other $Mg^{2+}$ channels (e.g., TRPM6/7)[5].

As all the studies to date have used measurements in cells[3–5,15,21–23], which preclude differentiation of direct or indirect mechanisms, it remains unclear whether CNNM/CorB proteins are themselves $Mg^{2+}$ transporters or regulators of other $Mg^{2+}$ transporters/channels.

Structurally, CNNM/CorB proteins are defined by a conserved core consisting of a transmembrane domain (TMD) and a cytosolic cystathionine-β-synthase (CBS)-pair domain (Fig. 1b)[24]. The TMD constitutes the largest family of protein domains of unknown function (DUF21) in the Pfam database[25]. The CBS-pair domains are composed of tandem CBS motifs and found in a wide variety of proteins, including the bacterial $Mg^{2+}$ channel MgtE, where they mediate dimerization and $Mg^{2+}$-ATP binding[26]. The CBS-pair domains of CNNMs have been characterized structurally and shown to undergo conformational changes upon $Mg^{2+}$-ATP binding[27,28]. Outside of the core, eukaryotic CNNM and prokaryotic CorB diverge. Eukaryotes have an extracellular domain and a cytosolic cyclic-nucleotide binding homology (CNBH) domain[29]. Prokaryotes lack the extracellular domain and have a distinct C-terminal CorC domain. Eukaryotic CNNMs contain an additional loop in the CBS-pair domain that allows binding of oncogenic phosphatase of regenerating liver (PRL) to regulate CNNM activities[30–32]. Only the soluble domains of CNNM/CorB have been characterized structurally until recently, during the preparation of this manuscript, the TMD fragment of a bacterial CorB ortholog (TpCorB) was reported[23].

Here, we determine the crystal structure of an archaeal CorB ortholog in two distinct conformations (apo and $Mg^{2+}$-ATP-bound). The TMD exists in an inward-facing conformation with a bound $Mg^{2+}$ ion coordinated by a conserved π-helix. Major structural rearrangements in the cytosolic domains are observed upon $Mg^{2+}$-ATP binding, suggesting a mechanism of $Mg^{2+}$-ATP sensing. We employ a liposome-based transport assay to demonstrate the direct $Mg^{2+}$ transport function of CorB proteins. Together, these structural and functional insights provide a framework for understanding the function of human CNNMs and associated diseases.

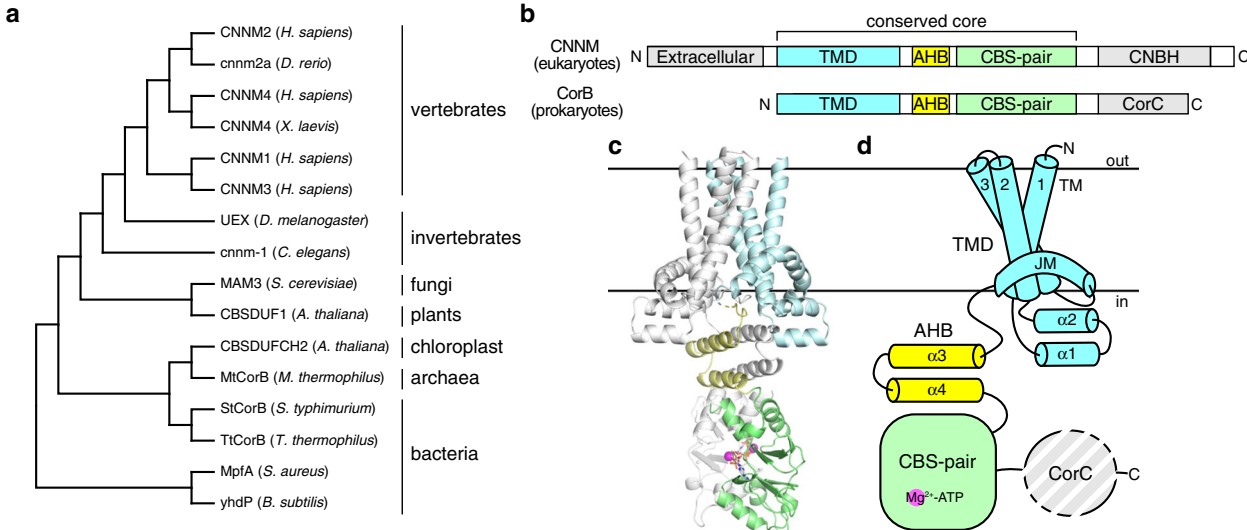

**Fig. 1 Overall structure and domain organization of CNNM/CorB $Mg^{2+}$ transporters. a** Phylogenetic analysis of representative CNNM/CorB orthologs generated using the neighbor-joining method. **b** Domain organization of eukaryotic CNNM and prokaryotic CorB. TMD transmembrane domain, AHB acidic helical bundle, CNBH cyclic nucleotide-binding homology domain, CorC cobalt resistance C domain. **c** Crystal structure of the $Mg^{2+}$-ATP-bound MtCorB without the C-terminal CorC domain as a homodimer. One chain is colored by domains. **d** Topology of a MtCorB monomer showing the transmembrane and juxtamembrane helices of the TMD (residues 1–154, cyan), the two helices of the AHB (residues 166–199, yellow), the $Mg^{2+}$-ATP-binding CBS-pair domain (residues 200–324, green), and the CorC domain (residues 325–426, grey).

## Results

**Structure determination**. To identify CNNM/CorB orthologs with suitable properties for structural studies, we performed small-scale screening of 20 prokaryotic CorB orthologs from diverse species (Supplementary Fig. 1). Each ortholog was expressed as a GFP-fusion protein and purified in six different detergents (DDM, LMNG, OGNG, LDAO, C12E9, and OG). Among them, two orthologs showed good expression and gel-filtration profiles: MtCorB from a thermophilic archaeon (*Methanoculleus thermophilus*) and TtCorB from a thermophilic, gram-negative bacterium (*Tepidiphilus thermophilus*). MtCorB and TtCorB share 26 and 25% sequence identities to the conserved core of human CNNM2, respectively (Supplementary Fig. 2). Attempts in crystallizing the full-length proteins yielded no suitable crystals. Thus, several deletion mutants were generated. MtCorB with a C-terminal CorC domain deletion (MtCorBΔC) purified in DDM yielded crystals in presence of $Mg^{2+}$-ATP. However, upon structural elucidation, the unit cell only contained the soluble CBS-pair domain (Table 1).

After extensive construct and detergent optimizations, we ultimately obtained diffracting crystals of MtCorBΔC with an internal loop deletion (MtCorBΔC$_{\Delta loop}$) purified in UDM in complex with $Mg^{2+}$-ATP (Supplementary Fig. 1d). The crystals were obtained by vapor diffusion in presence of 0.1 M sodium citrate pH 5.5, 0.1 M $Li_2SO_4$, 0.1 M NaCl, 20 mM $MgCl_2$, 34% PEG400 and 5 mM ATP. Initial phases were determined by molecular replacement using MtCNNM$_{CBS}$ structure solved earlier, which represented 40% of the crystallized construct (residues 199–322). Two α-helices (residues 166–198) preceding the CBS-pair domain were built-in manually, then PHENIX AutoBuild[33] was used to place idealized helical fragments in the missing sections (residues 1–166). The resulting phases were of great quality that allowed tracing of most of the molecule, guided by strong densities of bulky residues (Supplementary Fig. 3a). The final structure, consisting of chain A (residues 5–322) and chain B (residues 4–154 and 160–322), is refined to 3.25 Å with an *R*-free of 0.27 (Table 1).

**Architecture and domain organization**. MtCorBΔC crystallizes as a homodimer with each protomer consisting of three distinct regions: a TMD, an acidic helical bundle (AHB), and a CBS-pair domain (Fig. 1c, d). The TMD and the AHB represent previously unknown structural regions that required de novo model building. Since we were able to visualize the linker connecting TMD and AHB, we were able to assign the two chains with confidence, with the right TMD connecting to the right CBS-pair domain. Our result shows a discrepancy with the hypothetical model generated by combining isolated TMD and CBS-pair domain structures of bacterial TpCorB, in which the right TMD was connected to the left CBS-pair domain[23].

Although MtCorB is present as a dimer, it does not show C2 symmetry (Supplementary Fig. 3b). The individual TMD, AHB, and CBS-pair domains form symmetric dimers, but their arrangement is not. Overlaying the TMD of two polypeptide chains reveals asymmetry in the positions of the cytosolic domains, primarily due to differences in the linker between the TMD and AHB (Supplementary Fig. 3c). Together with previous observations that cytosolic domains of CNNMs form symmetric dimers[27–30,32,34], these results suggest the interface between the transmembrane and cytosolic domains is dynamic, and the current asymmetric arrangement likely arises from crystal packing (Supplementary Fig. 3d).

**TMD in an inward-facing conformation with a negatively charged cavity**. The TMD is a dimer with each chain composed of three transmembrane helices (TM1–3), a pair of short helices exposed on the intracellular side, and a juxtamembrane (JM) helix that forms a belt-like structure (Fig. 2a). The homodimeric

---

**Table 1 Statistics of data collection and refinement.**

| | MtCorB$_{CBS}$ + $Mg^{2+}$-ATP | MtCorBΔC$_{\Delta loop}$ + $Mg^{2+}$-ATP | MtCorBΔC R235L |
|---|---|---|---|
| *Data collection* | | | |
| X-ray source | ALS 5.0.2 | CLS 08ID-1 | CLS 08B1-1 |
| Wavelength (Å) | 1.00003 | 0.97996 | 1.52154 |
| Space group | P4$_1$22 | P2$_1$2$_1$2$_1$ | C2 |
| *Cell dimensions* | | | |
| *a, b, c* (Å) | 52.10, 52.10, 112.28 | 61.05, 118.68, 177.31 | 139.17, 124.02, 86.26 |
| *α, β, γ* (°) | 90.0, 90.0, 90.0 | 90.0, 90.0, 90.0 | 90.0, 92.8, 90.0 |
| Resolution (Å) | 50.00-2.05 (2.09-2.05)$^a$ | 50.00-3.25 (3.31-3.25)$^a$ | 50.00-3.80 (3.94-3.80)$^a$ |
| Redundancy | 22.0 (12.4) | 11.9 (9.1) | 6 (4.5) |
| Completeness (%) | 99.8 (97.3) | 99.2 (96.9) | 91.7 (78.7) |
| $I/\sigma I$ | 38.4 (1.4) | 22.0 (1.0) | 16.3 (0.8) |
| CC$_{1/2}$ | 0.992 (0.721) | 0.993 (0.430) | 0.965 (0.510) |
| *Refinement* | | | |
| Resolution (Å) | 38.19-2.05 | 49.31-3.25 | 46.27-3.80 |
| No. of reflections | 8835 | 14835 | 10821 |
| $R_{work}/R_{free}$ | 0.206/0.252 | 0.226/0.268 | 0.243/0.285 |
| *No. of atoms* | | | |
| Protein | 930 | 4786 | 4289 |
| Ligands | 37 | 299 | NA |
| Water | 56 | NA | NA |
| *B-factors* | | | |
| Protein | 35.1 | 27.5 | 95.9 |
| Ligands | 27.3 | 42.7 | NA |
| Water | 37.3 | NA | NA |
| *RMSDs* | | | |
| Bond lengths (Å) | 0.002 | 0.002 | 0.002 |
| Bond angles (°) | 0.54 | 0.50 | 0.44 |
| Ramachandran plots | | | |
| Favored (%) | 97.5 | 96.7 | 95.5 |
| Allowed (%) | 2.5 | 3.3 | 4.5 |
| Disallowed (%) | 0.0 | 0.0 | 0.0 |
| PDB code | 7MSU | 7M1T | 7M1U |

$^a$Highest resolution shell is shown in parentheses.

TMD exists in an inward-facing conformation with dimerization predominantly formed by hydrophobic contacts of TM2 and TM3 of each protomer. Electrostatic surface potential analysis reveals a large, negatively charged cavity that has a maximum diameter of approximately 10 Å and is lined with conserved polar residues (Fig. 2b, c and Supplementary Fig. 2). A well-defined electron density is observed in the cavity (Fig. 2d), and the density was modeled as an $Mg^{2+}$ ion based on the TpCorB structure[23]. The $Mg^{2+}$ is coordinated by hydroxyl groups of Ser21, Ser25, and Ser71; carboxyl group of Glu111; and the main-chain carbonyl groups of Ser21 and Gly110. Residues contributing to ion interactions are strongly conserved across species, suggesting these are conserved features of the CNNM/CorB proteins (Supplementary Fig. 2).

A second conserved feature is a π-helical turn in TM3, in which the helix is composed of an $i + 5$ instead of $i + 4$ configurations, resulting in a bulge in the helix (Fig. 2e and Supplementary Fig. 3a). The residues surrounding the helical turn are highly conserved with Glu111, Ile112, and Pro114 completely invariant from archaea to humans (Fig. 2f). Glu111 points toward the negative cavity and is involved in the coordination of the $Mg^{2+}$ ion, whereas Pro114 acts as a helix-breaker that allows the shift in hydrogen bonding of the π-helical turn. The positively charged Lys115 after Pro114 is also highly conserved across species.

The archaeal MtCorB shares sequence identify of 37% with the bacterial TpCorB. The bacterial TMD also exists as a homodimer in an inward-facing conformation with similar TM rearrangements (Supplementary Fig. 4a). Although it was not pointed out in the bacterial study[23], the bacterial structure also contains the acidic cavity and the conserved π-helical turn (Supplementary Fig. 4b, c). The polar residues in the acidic cavity and $Mg^{2+}$-binding site are highly conserved with all five $Mg^{2+}$-interacting residues being conserved except for Ser71, which is substituted by asparagine in bacteria.

**A re-entrant JM helix encircles TM helices.** The re-entrant JM helix following TM3 wraps around TM1 and TM2 like a belt through several hydrophobic contacts (Fig. 3a, b). The curvature is imposed by a conserved Pro142 that makes a kink in the helix (Supplementary Fig. 2). The outer surface of the JM helix is amphipathic and rich in aromatic residues. Surrounding the JM helix, we observed several electron densities, which we modeled as UDM detergent molecules from the protein buffer. In total, we observed ten UDM molecules, which likely correspond to conserved lipid-binding sites important for assisting conformational changes during transport. Structural comparison to the bacterial JM helix shows that the bacterial one is longer and makes more extensive contacts with TM3 of the other protomer

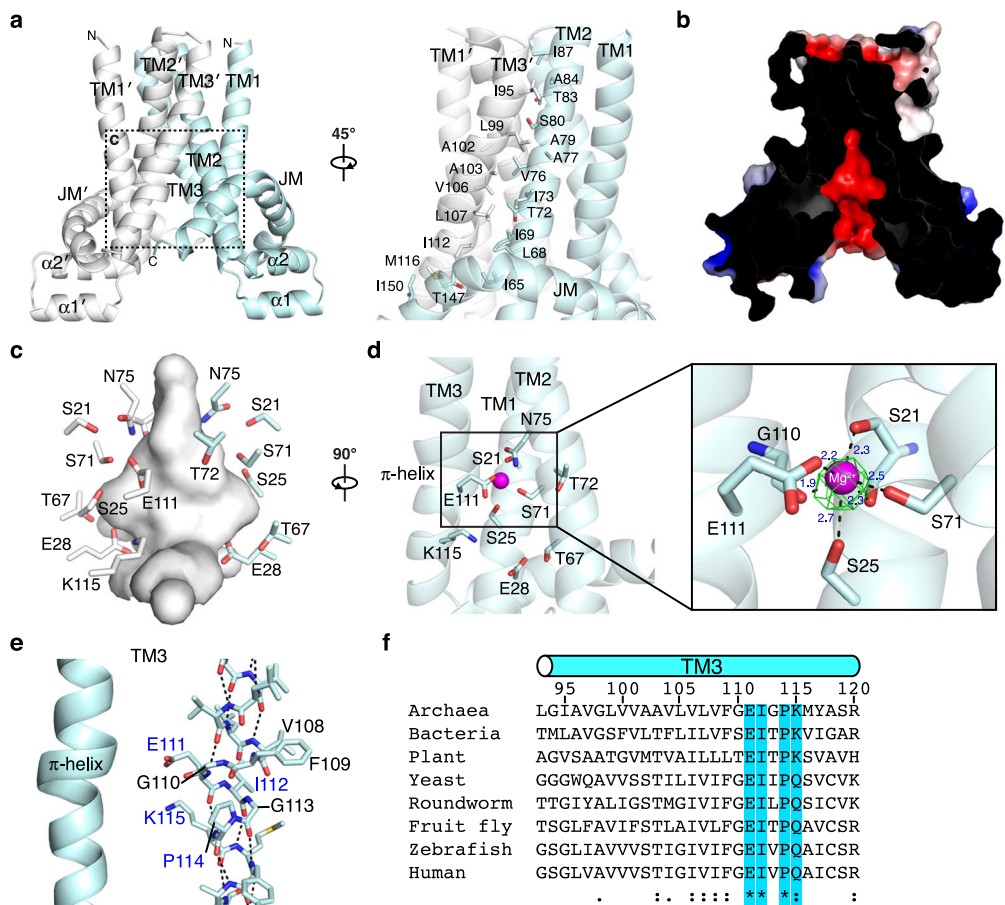

**Fig. 2 TM domain in an inward-facing conformation with a negatively charged cavity. a** TM domain homodimerizes with interface formed by TM2 and TM3 of each protomer. **b** Electrostatic surface potential representation (±5 kT e$^{-1}$) of MtCorB TM domain showing a cross-sectional view of the negatively charged cavity. **c** Close-up view of the polar residues forming the cavity. **d** A magnesium ion ($Mg^{2+}$) bound in the cavity with $F_o$–$F_c$ omits map contoured at 5.0σ. **e** π-helical turn preceding Pro114 in TM3. Highly conserved residues are highlighted in blue. **f** Conservation of residues in the π-helical turn from archaea to humans.

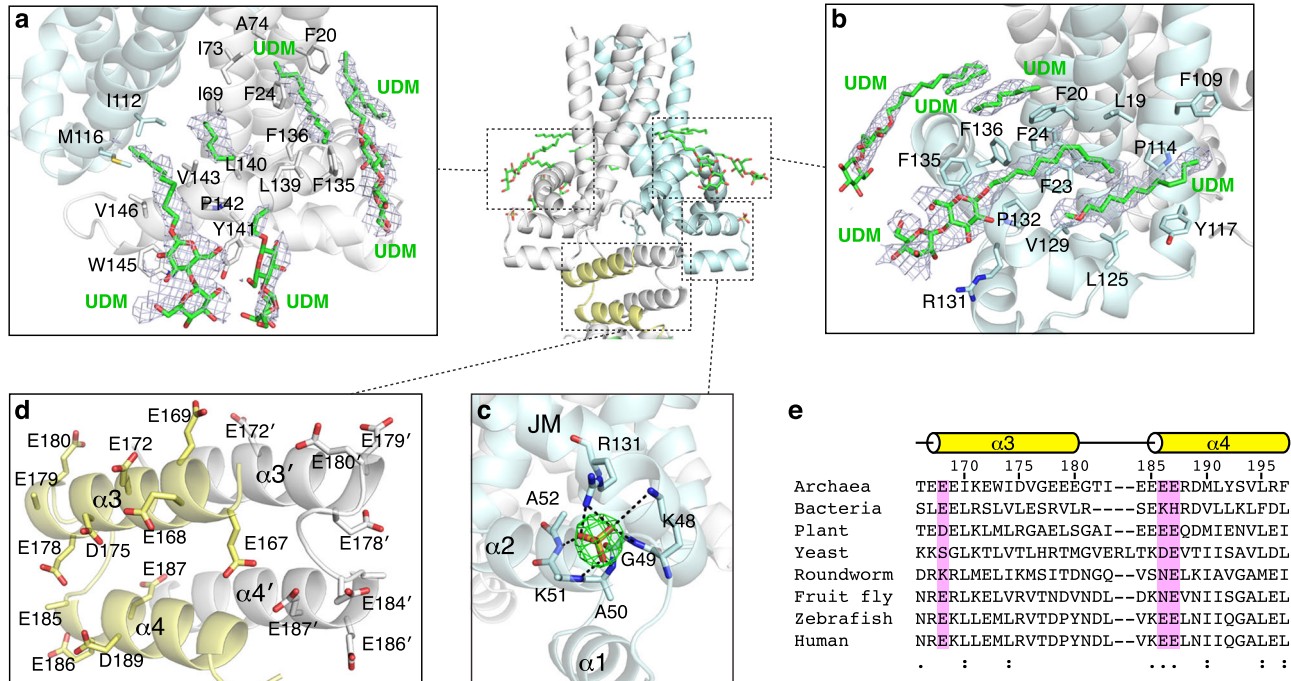

**Fig. 3 Juxtamembrane helix and acidic helical bundle. a, b** UDM detergent molecules bound by the juxtamembrane helix. Simulated annealing $2F_o$-$F_c$ composite omit map contoured at $1.0\sigma$. **c** A sulfate ion bound between JM and helix–turn–helix motif with $F_o$-$F_c$ omit map contoured at $5.0\sigma$. **d** Conservation of acidic residues in the AHB domain. **e** Sequence conservation of conserved glutamates in the AHB.

(Supplementary Fig. 4d). This discrepancy could be species-specific or caused by the absence of the following domains in the bacterial TMD structure[23].

Underneath the JM helix resides the helix-turn-helix motif (α1 and α2) that connects TM1 and TM2 (Fig. 3c). The solvent-exposed portions contain several basic residues, which may be involved in binding phospholipid headgroups. Supporting this, a strong density was observed in this region and assigned as a sulfate molecule due to its presence in the crystallization buffer.

**Acidic helical bundle**. An unexpected feature of the MtCorB structure is the existence of the four-helical AHB domain following the TMD (Fig. 3d). The linker between the JM helix and AHB differs in the two protomers. It is extended and partially disordered in one and coiled in the other. The AHB itself is a symmetric dimer composed of two helices from each protomer and highly negatively charged. Thirteen of the 31 residues are acidic with three stretches of three or more consecutive glutamic acids, which likely represent cation binding sites. The negative charge of the AHB is well conserved across evolution with the exception of yeast and roundworms (Fig. 3e).

**Structural basis of $Mg^{2+}$-ATP binding by the CBS-pair domain**. The isolated $MtCorB_{CBS}$ structure from the in situ cleavage of MtCorBΔC allowed us to visualize the detailed interactions of $Mg^{2+}$-ATP binding at higher resolution (Fig. 4a and Table 1). As observed in the structure with the TMD, two molecules of $Mg^{2+}$-ATP are bound to the central cavity of the CBS-pair dimer. The adenine bases are sandwiched between Phe233 and Ile311 in a hydrophobic pocket comprising Met298, Phe237, Ile236, Val212, and Val213. The ATP ribose forms hydrogen bonds with the side chains of Thr207, Asp316, and Thr313. The phosphate groups are stabilized with Arg235 and Ser234 with the bound $Mg^{2+}$ coordinated in an octahedral arrangement with three phosphates and three water molecules.

Measurements of the affinity of adenosine nucleotides by isothermal titration calorimetry (ITC) showed MtCorBΔC has the highest affinity for ATP, followed by ADP and AMP (Fig. 4b and Supplementary Fig. 5). While the addition of $Mg^{2+}$ has a relatively large effect on the binding of ATP to the isolated CBS-pair domain in human CNNMs[27,35], the effect was more muted with MtCorB. In presence of $Mg^{2+}$, the affinity for ATP was increased three-fold with MtCorBΔC and ten-fold with the isolated $MtCorB_{CBS}$ domain (Supplementary Fig. 5). These affinities are comparable to those measured from the isolated CBS-pair domains of TpCorB[23].

We used analytical ultracentrifugation (AUC) to characterize the effect of ATP binding on MtCorB (Fig. 4c and Supplementary Fig. 6). As observed for human CNNMs[27], dimerization of the CBS-pair domain was tightly coupled with $Mg^{2+}$-ATP binding. While the $MtCorB_{CBS}$ in the absence of ATP sedimented as a monomer, the addition of adenine nucleotides triggered dimerization. The addition of $Mg^{2+}$ alone had no effect on the sedimentation of $MtCorB_{CBS}$.

Next, we turned to hydrogen–deuterium exchange mass spectrometry (HDX-MS) to detect conformational changes in full-length MtCorB upon ATP binding and deletion of the CorC domain. MtCorB and MtCorBΔC were first reconstituted into MSP1D1 nanodiscs (Supplementary Fig. 7a), and hydrogen exchange was monitored in the presence and absence of $Mg^{2+}$-ATP. While peptide coverage of the hydrophobic TMD was limited, coverage of the cytosolic domains was excellent and revealed large conformational changes (Fig. 4d and Supplementary Fig. 7b, c). The addition of $Mg^{2+}$-ATP significantly reduced the exchange in the CBS-pair domain dimerization interface and ATP-binding site. Similar changes were observed with both the full-length and truncated protein, which is consistent with the absence of a role of the C-terminal domain in $Mg^{2+}$-ATP binding. This is different from human CNNMs where the C-terminal CNBH domain promotes CBS-pair domain dimerization and $Mg^{2+}$-ATP binding[27]. Comparison of full-length and

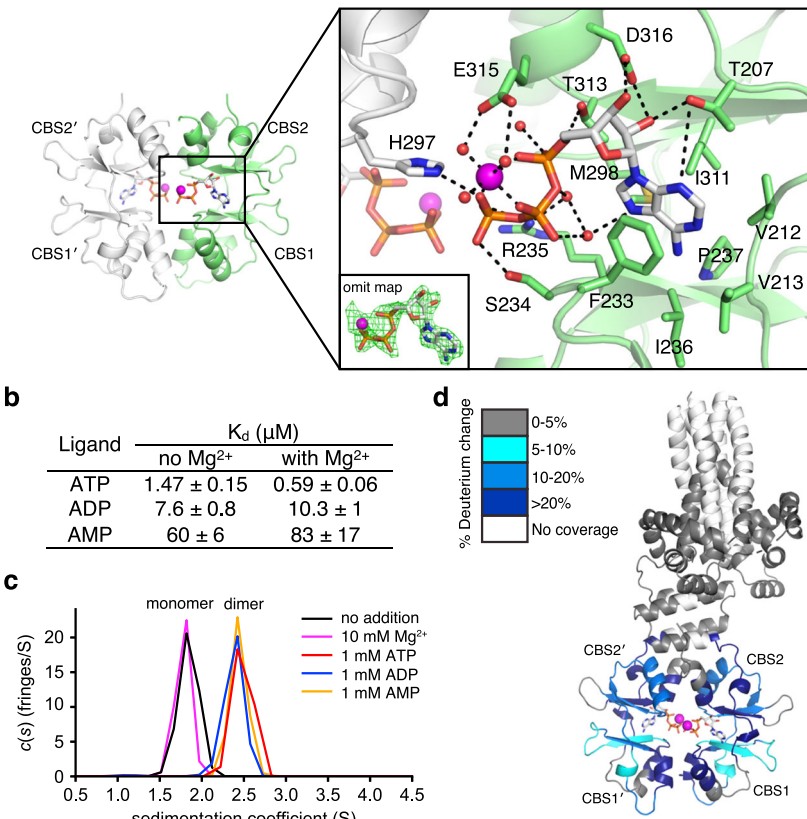

**Fig. 4 Mg²⁺-ATP binding to CBS-pair domain. a** Structural basis of Mg²⁺-ATP binding. Mg²⁺ ions and water molecules are shown in *magenta* and *red*, respectively. The Mg²⁺-ATP $F_o$–$F_c$ omits map was contoured at 3.0σ. **b** Affinities of MtCorBΔC to adenosine nucleotides with and without 50 mM Mg²⁺ measured by ITC. **c** Dimerization of the MtCorB CBS-pair domain in the presence of 1 mM adenosine nucleotides as measured by SV-AUC experiments. **d** Conformational change in the MtCorBΔC in the presence and absence of Mg²⁺-ATP measured by HDX-MS. Regions that showed significant decreases in exchange (defined as >5%, 0.4 Da, and a two-tailed Student's *t* test *p* < 0.01) in the presence of Mg²⁺-ATP are colored blue. Peptides in the CBS-pair domain dimerization interface show the most protection from deuterium exchange upon Mg²⁺-ATP binding.

truncated MtCorB showed changes in the CBS2 motif, suggesting this is the primary site of interaction of the CorC domain.

**Major structural rearrangements in MtCorB upon loss of Mg²⁺-ATP binding**. Further insight into the conformation in the absence of bound ATP came from the crystal structure of a MtCorBΔC R235L mutant, homologous to a Jalili syndrome mutant (R407L) that is unable to bind ATP in human CNNM4[27]. Crystals of MtCorBΔC R235L mutant purified in UDM were obtained by vapor diffusion in presence of 0.1 M sodium citrate pH 5.5, 50 mM Li₂SO₄, and 11% PEG4000, but they diffracted poorly to 6 Å. Subsequent additives screening improved the resolution to 3.8 Å with the addition of 10 mM MgSO₄ (Table 1). The structure was determined by molecular replacement using individual TM and CBS-pair domains, while the AHB was rebuilt manually.

In the apo structure, the TMD remains in the same inward-facing conformation, but the cytosolic domains undergo major structural rearrangements (Fig. 5a). The CBS-pair domain has completely dissociated from the disc-like dimeric configuration to a more elongated, open conformation with different domain–domain contacts. To our knowledge, this dimeric configuration is not observed in any available CBS-pair domain structures[26]. In agreement with the AUC results, the loss of ATP binding disrupted the dimerization of the CBS-pair domain, which then makes contact with the helix-turn-helix motif from TMD (Fig. 5b). In addition, the dimeric AHB dissociates and binds to

the CBS2 motif of the neighboring molecule, competing for the Mg²⁺-ATP binding site (Fig. 5c).

The morphing of the two conformations allowed us to visualize the conversion of the apo to the Mg²⁺-ATP bound form (Supplementary Movie 1). The two conformations are interchangeable by a 90° rotation of both protomers (Fig. 5d). Despite the large dimeric rearrangement, the individual protomer in the two states is similar with differences limited to the AHB and α8 helix (Fig. 5e). Our result shows a discrepancy to the isolated CBS-pair domain structures of the bacterial TpCorB, in which minimal structural changes were observed upon nucleotide binding[23]. Perhaps, this was due to the absence of TMD and one of the AHB helices in the bacterial structures.

**Liposome-based Mg²⁺ transport assays**. In order to address the most debated question in the field: whether CNNM/CorB proteins mediate direct or indirect Mg²⁺ transport, we carried out an in vitro, liposome-based Mg²⁺ transport assay using a modified version of assays used for CorA proteins[36,37]. We reconstituted CorB proteins into liposomes and encapsulated mag-fura-2, a ratiometric Mg²⁺ indicator (Fig. 6a). Upon addition of Mg²⁺, we observed a time-dependent change in the ratio of fluorescence from excitation at two wavelengths (330/369 nm), which could be used to calculate the Mg²⁺ concentration inside the liposomes. CorB liposomes showed a time-dependent increase in the fluorescence ratio while no fluorescence change was observed for liposomes devoid of protein. Assays with the bacterial CorB

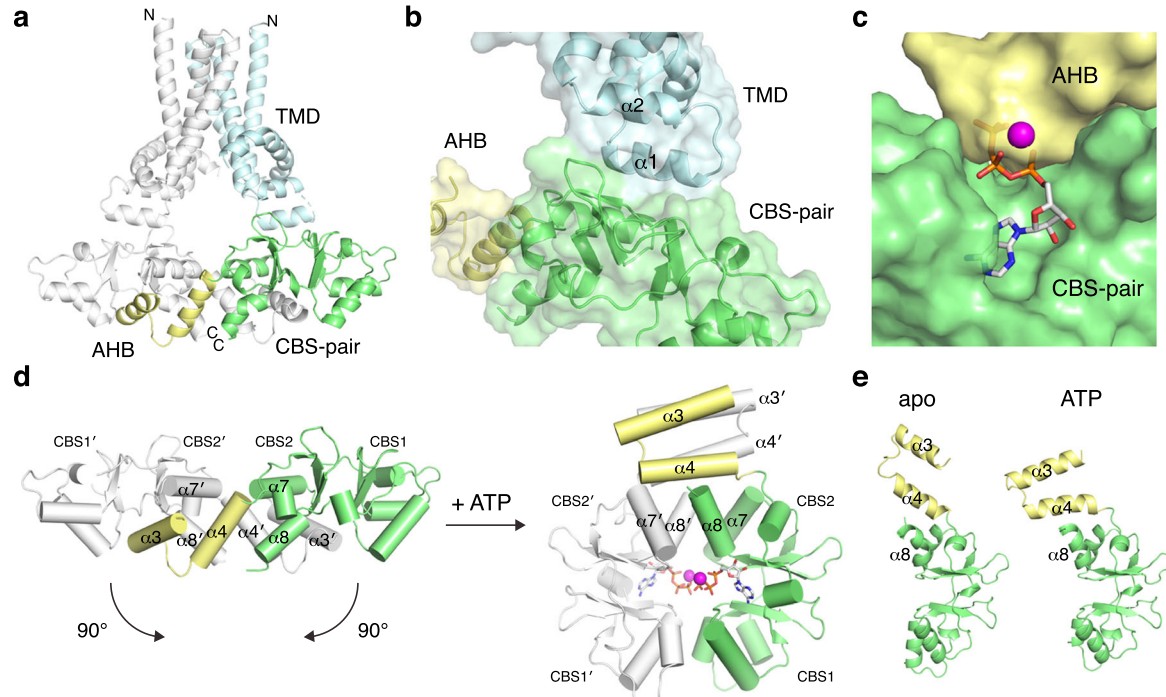

**Fig. 5 Apo conformation of MtCorBΔC captured by R235L mutant. a** Overall structure of MtCorBΔC R235L mutant. One chain is colored by domains. **b** Novel domain–domain contact between TMD and CBS-pair domain. **c** AHB binds to CBS-pair domain and competes for $Mg^{2+}$-ATP site. **d** Large conformational change in the AHB and CBS-pair domains upon $Mg^{2+}$-ATP binding. **e** The apo and ATP-bound structures are similar with differences limited to the AHB and α8 helix.

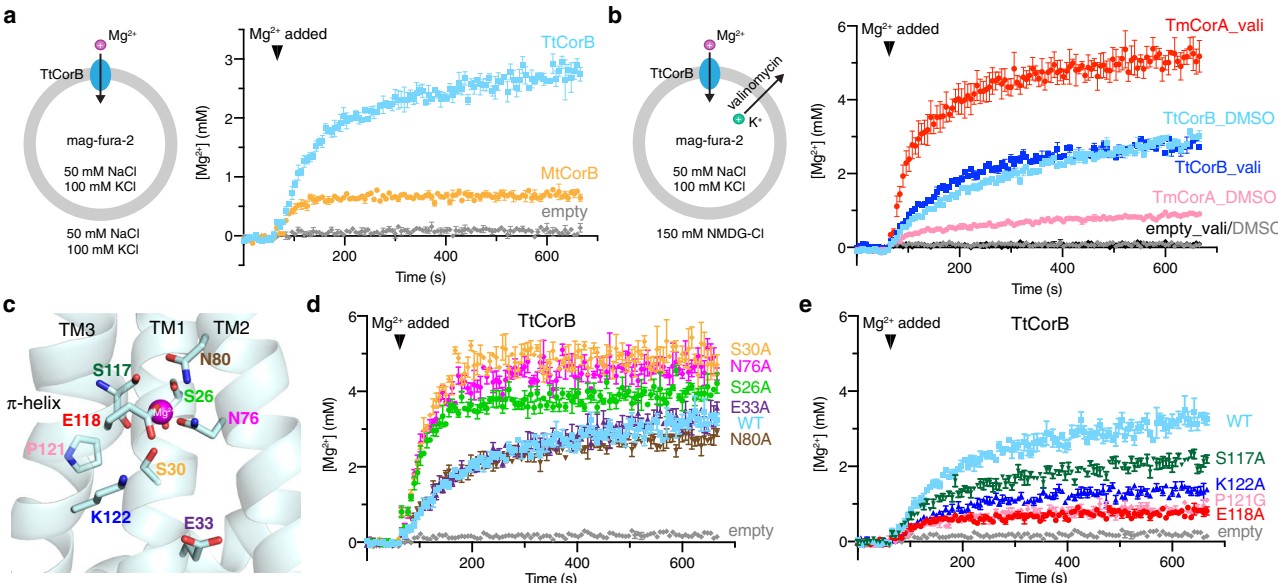

**Fig. 6 Liposome-based assay shows direct $Mg^{2+}$ transport by CorB proteins. a** Liposomal assay showing direct transport of $Mg^{2+}$ by MtCorB and TtCorB. Proteoliposomes (protein/lipid ratio of 1:30) or empty liposomes containing mag-fura-2 were equilibrated for 1 min before the addition of 5 mM $MgCl_2$ to initiate $Mg^{2+}$ uptake (*arrowhead*). The data points represent mean ± SEM ($n = 3$ independent measurements). **b** An inward negative membrane potential generated by valinomycin leads to enhanced $Mg^{2+}$ transport by TmCorA (protein/lipid ratio of 1:270) but not TtCorB (protein/lipid ratio of 1:30), suggesting electroneutral transport of $Mg^{2+}$ by TtCorB. The data points represent mean ± SEM ($n = 3$ independent measurements). **c** Homology model of TtCorB showing locations of conserved residues in the negatively charged cavity and $Mg^{2+}$ binding site. **d, e** Liposomal $Mg^{2+}$ transport assays of TtCorB mutants under the same experimental conditions as panel (**a**). Mutations in the $Mg^{2+}$-binding site increased transport activity while mutations near the π-helical turn reduced transport. The data points represent mean ± SEM ($n = 3$ independent measurements).

(TtCorB) showed more transport activity than the archaeal protein (MtCorB), possibly due to better compatibility with the lipid composition of the proteoliposomes. Thus, we used the bacterial protein for further functional studies. By varying the protein/lipid ratio and the concentration of $Mg^{2+}$, we showed that the TtCorB-mediated $Mg^{2+}$ transport is protein- and magnesium-dependent (Supplementary Fig. 8a, b).

Next, to test whether the TtCorB-mediated $Mg^{2+}$ transport is electrogenic, we used valinomycin, a potassium-selective ionophore, to generate an inward negative membrane potential (Fig. 6b). We used TmCorA, a well-studied $Mg^{2+}$ channel, as a positive control. As shown previously, TmCorA showed increased $Mg^{2+}$ accumulation in presence of valinomycin[36,37]. In contrast, we did not observe an increase in $Mg^{2+}$ transport for TtCorB. These results suggest TtCorB mediates $Mg^{2+}$ transport through an electroneutral antiporter mechanism. We also examined the ion selectivity of TtCorB toward different divalent cations. Due to the differences in the affinity and response of mag-fura-2 to different ions, it is difficult to directly compare ions, but we could compare the ion selectivity of TtCorB with TmCorA. In our experimental conditions, TtCorB was more selective and showed less transport of $Ca^{2+}$ and $Zn^{2+}$ than TmCorA (Supplementary Fig. 8e). TtCorB and TmCorA showed similar transport activity with $Mg^{2+}$.

Lastly, to probe the importance of the conserved residues lining the negatively charged cavity in TMD (Fig. 6c), we tested the effect of several cavity mutations on the $Mg^{2+}$ transport activity. During purification, some mutants, especially ones involved in $Mg^{2+}$ coordination, have abnormal gel-filtration profiles compared to wild-type proteins as shown by the earlier elution volumes (Supplementary Fig. 8c), suggesting the importance of these residues in maintaining the structural integrity of the protein. Despite that, all mutants showed high purity and incorporated well into liposomes (Supplementary Fig. 8d). $Mg^{2+}$ transport assays were performed on all the mutants (Fig. 6d, e). Some mutants (E33A and N80A) have near-wildtype transport activity, while others, such as the $Mg^{2+}$-binding site mutants (S26A, S30A, and N76A), have increased transport activity. On the other hand, mutations of the four conserved residues (S117A, E118A, P121G, and K122A) near the π-helical turn consistently reduced the $Mg^{2+}$ transport activity. These results demonstrate the functional importance of the conserved residues in the $Mg^{2+}$-binding site and π-helical turn.

Huang et al.[23] had also tested the effect of several cavity mutations with an indirect, cell-based $Mg^{2+}$ efflux assay through the expression of chimeric CNNM4/TpCorB proteins in HEK293 cells. In comparison, we have similar results for S117A and E118A equivalent mutants, which showed reduced $Mg^{2+}$ efflux activity. In contrast, divergent results were obtained for S30A and E80A equivalent mutants. Both mutants had reduced efflux activity in the bacterial TpCorB, while our S30A mutant increased activity, and the N80A mutant has near-wildtype activity. Huang et al. did not measure transport activity with mutations equivalent to S26A, E33A, N76A, P121G, and K122A.

**Homology modeling of human CNNM.** We used homology modeling to improve our understanding of mutations in human CNNM proteins (HsCNNMs) that cause diseases. Previous attempts to predict the structure of HsCNNM2 were confounded by the hydrophobicity of the HsCNNM2 JM helix[38] (Supplementary Fig. 9a). Using the $Mg^{2+}$-ATP bound structure of MtCorBΔC, we generated a homology model of HsCNNM2, which showed good geometry and allowed TM helices to be reliably positioned (Fig. 7a and Supplementary Fig. 9). The main differences lie in the length of linkers connecting the TM helices.

The HsCNNM2 model reproduced the negatively charged cavity, π-helix in TM3, the $Mg^{2+}$ ion binding site, and the presence of buried polar residues (e.g., Thr331, Asp335, and Ser348), supporting the importance of these features for CNNM function (Fig. 7b, c).

We used the model to understand the molecular basis for human disease mutations (Fig. 7d). Nine missense mutations in HsCNNM2 are responsible for hypomagnesemia[3,5,39,40]. T568I in the CBS-pair domain prevents ATP binding and $Mg^{2+}$ transport[27,35]. L418P in the dimerization interface of AHB likely disrupts the helical bundle. In the TMD, S269W, and L330F both likely disrupt TM helix packing, and E357K is a charge reversal mutation of the invariant glutamate that coordinates the $Mg^{2+}$. D335N near the extracellular surface may be important in the outward-facing conformation. The high degree of sequence conservation (84% identity) between HsCNNM2 and HsCNNM4 also allowed us to use the model to interpret HsCNNM4 mutations that cause Jalili syndrome[41–46]. Of particular interest are mutations that occur in the TMD. S196P and S200Y both map to the negatively charged cavity. HsCNNM4 S196P affects the same residue as CNNM2 S269W, confirming the importance of this position for function. R236Q, S245L, and L324P are located at the inner surface of the membrane, in TM3, and the JM helix, respectively.

**Molecular dynamics simulations.** We used molecular dynamics simulations to study asymmetry and conformational changes in the MtCorBΔC structure. Unbiased MD simulations of the MtCorBΔC structure were first carried out (Supplementary Movie 2). As expected from the asymmetry in the structure, the interface between the TMD and cytosolic domains was flexible and large movements were observed. At the end of the simulation, the conformations of the two protomers were essentially switched while the individual domains all retained their symmetry. In addition, we also performed unbiased MD simulations of symmetric MtCorBΔC dimer (Supplementary Movie 3). In this case, the cytosolic domains appear more rigid than that of asymmetric dimer. We also observed an upward movement of the AHB towards the TMD, suggesting entrapment and stabilization by the helix-turn-helix motif of TMD.

The current structure resembles a transporter fold in the inward-facing conformation as supported by the hydration analysis of the structure (Supplementary Fig. 10a). In order to simulate the outward-facing conformation, we pulled an $Mg^{2+}$ ion through the TMD dimerization interface while keeping a distance constraint with Glu111 of both protomers. This in turn causes the two helices to flip open, generating a possible outward-facing conformation (Fig. 7e). We then performed targeted MD to gain insight into the conformational changes from the inward to the outward-facing conformation (Supplementary Movie 4 and Supplementary Fig. 10b, c). The conserved residues Glu10, Ser80, and Thr83, which are buried in the membrane in the inward-facing conformation, become solvent-exposed and available for ion binding. The conservation of the π-helix and buried polar residues suggest that the mechanism of ion transport is conserved across all CNNM/CorB family members.

## Discussion

$Mg^{2+}$ is the most abundant divalent cation inside cells and essential for a wide variety of biochemical and enzymatic reactions[1]. Despite this, our structural knowledge of $Mg^{2+}$ ion transporter is limited to three proteins: MgtE[47], CorA[48–50], and TRPM7[51]. The structure of CorB proteins defines a different class of $Mg^{2+}$ transporter, highly conserved and present in all organisms.

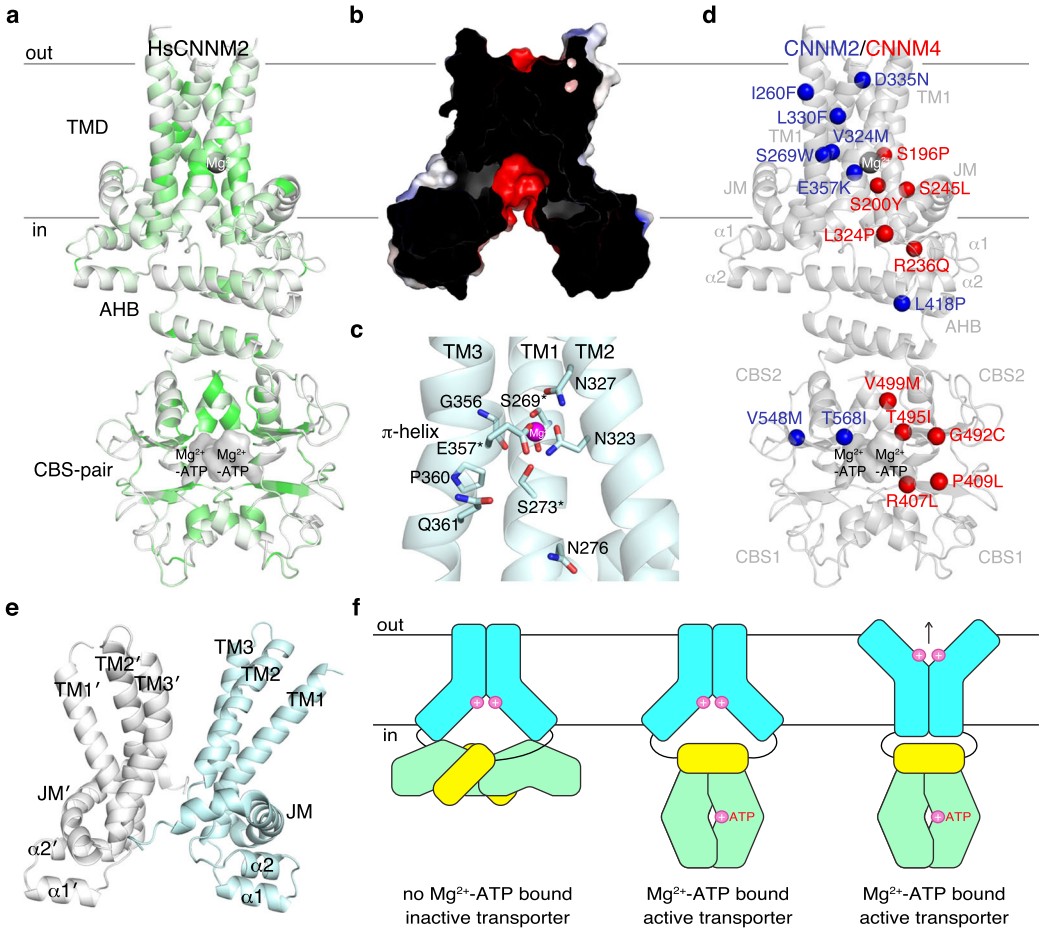

**Fig. 7 Molecular modeling. a** Homology model of human CNNM2 (HsCNNM2) color-coded according to identity to CNNM/CorB proteins shown in Supplementary Fig. 2. Conservation is highest around the $Mg^{2+}$ binding site, negatively charged cavity, and the CBS-pair dimerization interface. **b** Electrostatic surface potential analysis ($\pm 5\,kT\,e^{-1}$) shows conservation of negatively charged cavity in HsCNNM2 in cross-sectional view. **c** Conservation of the polar residues in the $Mg^{2+}$ binding site and $\pi$-helical turn of HsCNNM2. Asterisk denotes disease-causing mutations. **d** Mutations in CNNM2 and CNNM4 responsible for hypomagnesemia and Jalili syndrome cluster around the core of the TMD and nucleotide-binding site. For clarity, mutations are only shown on one chain. **e** Outward-facing conformation generated by MD simulations. **f** Proposed model of MtCorB transport and regulation. Loss of $Mg^{2+}$-ATP binding, CBS-TMD contact inactivates TMD in the inward-facing conformation. Upon $Mg^{2+}$-ATP binding, CBS-TMD contact is alleviated, thereby activating and allowing TMD to mediate $Mg^{2+}$ transport.

CNNM/CorB proteins had previously been suggested to be too small in ion transport due to the small number of predicted TM helices[18]. CorA is a pentamer with a total of 10 TM helices, MgtE is a dimer with 10, and TRPM7 has 24. Our structure shows that 6 TM helices suffice to form a large cavity (~1500 Å$^3$) with ion binding sites deep within the membrane (Fig. 2). In fact, the current structure resembles a transporter fold in an inward-facing conformation with solvent accessible cavity toward the cytosol. This suggests that CNNM/CorB proteins might transport $Mg^{2+}$ through a rocker-switch mechanism, in which the protein may exist in other conformational states, such as the occluded or outward-facing conformations. We have attempted to simulate the outward-facing conformation (Fig. 7e), but further studies are needed to validate and capture CNNM/CorB proteins in other conformational states.

The biggest controversy in the field surrounds the question: whether CNNMs/CorBs are $Mg^{2+}$ transporters themselves or $Mg^{2+}$ sensors that regulate other transporters? The dispute arises because all transport assays performed so far were done through overexpression of CNNMs/CorBs in cells[3–5,15,21–23], which preclude the differentiation of direct or indirect $Mg^{2+}$ transport. With our in vitro, liposome-based assay, we demonstrated direct

$Mg^{2+}$ transport of CorB proteins (Fig. 6). In addition, we showed that the transport likely occurs through an electroneutral antiporter mechanism, similar to that proposed in CNNMs[4]. Together with the structural insights, our results clearly demonstrate CNNM/CorB proteins are magnesium transporters themselves.

Throughout evolution, the CBS-pair domain of CNNM/CorB proteins remains associated with the TMD. The ATP-binding site is a disease mutational hotspot (Fig. 7d), and disruption of ATP-binding results in loss of $Mg^{2+}$ efflux activity[23,27,35]. While the CBS-pair domain has no ATPase activity[23,27], ATP-binding likely plays a regulatory role as was observed for the bacterial $Mg^{2+}$ channel MgtE[52]. Using a disease-mimicking mutant, we were fortunate to capture the apo state of the transporter (Fig. 5). Structurally, the loss of $Mg^{2+}$-ATP binding disrupts the dimerization of both CBS-pair and AHB domains. Dramatic changes in the dimeric rearrangement of CBS-pair domains upon nucleotide-binding have been well-documented in other CBS domain-containing proteins[26] as well as human CNNMs[27,28]. However, the current dimeric configuration involving AHB contact has not been observed before and seems CNNM/CorB specific. The acidic residues in the AHB are reminiscent of the acidic patches in the cytosolic domains of CorA and MgtE that

are involved in Mg$^{2+}$ sensing and regulation of channel activity[53,54]. Disruption of the AHB dimer could play a regulatory role through the loss of Mg$^{2+}$ binding sites. In addition, the CBS-TMD contact could result in an inactivation mechanism (Fig. 7f). For example, upon loss of ATP binding, the generation of the CBS-TMD contact would lock the TMD in the inward-facing conformation. This in turn inactivates the transporter as it is unable to change to other conformational states required for Mg$^{2+}$ transport. Upon Mg$^{2+}$-ATP binding, dimerization of the CBS-pair domain results in loss of CBS-TMD contact, and this, in turn, alleviates the inactivation of TMD, thereby allowing Mg$^{2+}$ transport to occur.

The CNNM/CorB family of proteins is highly homologous in the conserved core from humans down to bacteria. Despite their common origin, eukaryotic CNNMs have evolved to be different. In animals, PRL phosphatases provide an additional level of regulation through binding the CNNM CBS-pair domain. PRLs are highly oncogenic and promote cancer metastasis through their suppression of CNNM Mg$^{2+}$ efflux[9,10]. PRL binding itself is regulated by PRL phosphorylation in response to Mg$^{2+}$ levels[30,31]. Although further studies are needed to fully understand the function of human CNNMs and their regulation by PRLs, the structural and functional insights from this study provide a framework for understanding CNNM function and the development of therapeutic approaches for regulating their activity.

## Methods

### Construction of phylogenetic tree
Amino acid sequences of various CNNM/CorB orthologs were aligned using MUSCLE[55]. The phylogenetic tree was generated using the neighbor-joining method and bootstrapping of 1000 replications in MEGAX (Version 10.1.8)[56]. The CNNM/CorB orthologs and their UniProt accession numbers are: CNNM2 (*Homo sapiens*; Q9H8M5), cnnm2a (*Danio rerio*; A2ATX7), CNNM4 (*H. sapiens*; Q6P4Q7), CNNM4 (*Xenopus tropicalis*; A0JPA0), CNNM1 (*H. sapiens*; Q9NRU3), CNNM3 (*H. sapiens*; Q8NE01), UEX (*Drosophila melanogaster*; A0A0B7P9G0), cnnm-1 (*Caenorhabditis elegans*; A3QM97), MAM3 (*Saccharomyces cerevisiae*; Q12296), CBSDUF1 (*Arabidopsis thaliana*; Q67XQ0), CBSDUFCH2 (*Arabidopsis thaliana*; Q84R21), MtCorB (*M. thermophilus*; A0A1G8XA46), StCorB (*Salmonella typhimurium*; Q8XFY3), TtCorB (*T. thermophilus*; A0A0K6IWT9), MpfA (*S. aureus*; A0A0H3JL60), and yhdP (*B. subtilis*; O07585).

### Cloning of CorB orthologs
Codon-optimized cDNA of 20 CorB orthologs were synthesized (Bio Basic Inc., Markham, Canada) and sub-cloned into NcoI and XhoI sites of pCGFP-BC vector[57] with a C-terminal GFP-His8-tag for small-scale expression and detergent screening. Promising orthologs were subcloned into NdeI and XhoI sites of pET29a vector (Millipore Sigma) with a C-terminal His6-tag for large-scale expression. Constructs for *M. thermophilus* CorB (UniProt entry A0A1G8XA46): MtCorB (residues 1–426), MtCorBΔC (residues 1–322), MtCorBΔC$_{\Delta loop}$ (residues 1–322 Δ259–262), and MtCorBΔC R235L. Construct for *T. thermophilus* CorB (UniProt entry A0A0K6IWT9): TtCorB (residues 1–445). For MtCorB constructs lacking TMD, MtCorB$_{CBS+CorC}$ (residues 199–417), MtCorB$_{CBS}$ (residues 199–322), and MtCorB$_{CorC}$ (residues 333–417) were subcloned into BamHI and XhoI sites of pGEX-6P-1 vector (GE Healthcare) with an N-terminal GST-tag. QuikChange Lightning Site-Directed Mutagenesis Kit (Agilent) was used for the generation of point mutations. A list of mutagenic primers is included in Supplementary Table 1.

### Small-scale expression and detergent screening of CorB orthologs
CorB orthologs cloned in pCGFP-BC were transformed into *Escherichia coli* strain C41 (DE3). Cells were grown in Luria Broth (LB) at 37 °C to an optical density of 0.6 and induced with 1 mM IPTG for 4 h at 30 °C. The cell pellet was obtained by centrifuging at 5000*g* for 10 min. The pellet was re-suspended in lysis buffer (50 mM HEPES, 500 mM NaCl, 5% glycerol, pH 7.5) supplemented with cOmplete™ protease inhibitor cocktail (Roche) and split into 6 fractions. Lysis was performed using a 24-probe sonicator. Each fraction was solubilized with a different detergent (DDM, *n*-dodecyl-β-D-maltopyranoside; LMNG, lauryl maltose neopentyl glycol; OGNG, octyl glucose neopentyl glycol; LDAO, lauryldimethylamine-N-oxide; C12E9, dodecyl nonaethylene glycol ether; OG, *n*-octyl-β-D-glucopyranoside) to a final concentration of 1%, then purified by IMAC in the same detergent (3× CMC). Elutions were analyzed by sodium dodecyl sulfate-polyacrylamide gel electrophoresis (SDS-PAGE) and size-exclusion chromatography on SEPAX Zenic-C

SEC-300 connected to fluorescence detector (Ex 480 nm/ Em 510 nm) in buffer (50 mM HEPES pH 7.5, 150 mM NaCl, 0.5 mM TCEP, 3× CMC of detergent).

### Expression and purification of MtCorB, TtCorB, and TmCorA
MtCorB and TtCorB constructs were transformed into *E. coli* strain C41 (DE3), while TmCorA[50] was transformed into *E. coli* strain BL21 (DE3). MtCorB was grown in LB at 37 °C to an optical density of 0.6 and induced with 0.5 mM IPTG overnight at 18 °C. TtCorB and TmCorA were grown in LB at 37 °C to an optical density of 0.8 and induced with 0.5 mM IPTG for 4 h at 30 °C. The cell pellet was obtained by centrifuging at 5000*g* for 20 min. The pellet was re-suspended in lysis buffer (50 mM HEPES, 500 mM NaCl, 5% glycerol, pH 7.5) supplemented with cOmplete™ protease inhibitor cocktail (Roche), 10 μg/mL DNAse I, 1 mM CaCl$_2$, and 0.1 mg/mL lysozyme. MtCorB cells were lysed by passing through Avestin Emulsiflex-C3 homogenizer (10,000–15,000 p.s.i.), while TtCorB and TmCorA cells were lysed by sonication. Cellular debris was removed by centrifugation at 27,000*g* for 10 min at 4 °C (this step is omitted for full-length MtCorB). Membranes were pelleted by ultracentrifugation at 150,000*g* for 1 h at 4 °C. The membrane fraction was collected, flash-frozen in liquid nitrogen, and stored at −80 °C for later use. The membrane fraction after thawing was solubilized in lysis buffer supplemented with 1% DDM (MtCorB & TmCorA) or UDM (TtCorB) for 1 h at 4 °C on a rotator, then ultra-centrifuged at 150,000*g* for 30 min at 4 °C. The supernatant was loaded onto Qiagen Ni-NTA resin by batch binding, and incubated with gentle shaking for 1 h at 4 °C. The resin was then washed with wash buffer (50 mM HEPES, 500 mM NaCl, 5% glycerol, 30 mM imidazole, pH 7.5) containing 0.03% DDM or 0.05% UDM and eluted with elution buffer (50 mM HEPES, 500 mM NaCl, 5% glycerol, 300 mM imidazole, pH 7.5) containing 0.03% DDM or 0.05% UDM. The affinity-purified protein was further purified by size exclusion chromatography on a HiLoad 16/600 Superdex 200 pg column (GE Healthcare) in HPLC buffer (20 mM HEPES, 150 mM NaCl, pH 7.5) containing 0.03% DDM or 0.05% UDM. For TtCorB and TmCorA, the purification or HPLC buffers additionally contain 5 mM BME or 3 mM TCEP. The final purified proteins were concentrated using 50 kDa (MtCorBΔC) or 100 kDa (MtCorB, TtCorB, and TmCorA) cutoff concentrators (Amicon Ultra, Millipore), flash-frozen in liquid nitrogen, and stored at −80 °C for later use. The protein concentration is determined spectrophotometrically using Nanodrop, and purity is verified by SDS-PAGE.

### Expression and purification of MtCorB constructs lacking TMD
Constructs were transformed into *E. coli* strain BL21 (DE3). Cells were grown in LB at 37 °C to an optical density of 0.8 and induced with 1 mM IPTG overnight at 20 °C. The cell pellet was obtained by centrifuging at 5000*g* for 20 min. The pellet was re-suspended in lysis buffer (50 mM HEPES, 500 mM NaCl, 5% glycerol, pH 7.5) supplemented with cOmplete™ protease inhibitor cocktail (Roche), 10 μg/mL DNAse I, 1 mM CaCl$_2$, and 0.1 mg/mL lysozyme. Cells were lysed by sonication. Cellular debris was removed by centrifugation at 44,000*g* for 45 min at 4 °C. The supernatant was loaded onto Glutathione Sepharose resin (GE Healthcare), washed with lysis buffer, and eluted with lysis buffer containing 20 mM glutathione. The GST-tag was removed by overnight incubation with PreScission Protease, leaving an N-terminal Gly-Pro-Leu-Gly-Ser extension. The protein was further purified by size exclusion chromatography on a HiLoad 16/600 Superdex 75 pg column (GE Healthcare) in HPLC buffer (20 mM HEPES, 100 mM NaCl, pH 7.5). The protein was diluted to 5 μM, dialyzed overnight in dialysis buffer (20 mM HEPES, 100 mM NaCl, 5 mM EDTA, pH 7.5), and re-injected onto Superdex-75 to remove bound nucleotides. The final purified protein was concentrated to around 10 mg/mL (measured by NanoDrop), and the purity verified by SDS-PAGE.

### Crystallization
Crystals of MtCorB$_{CBS}$ with Mg$^{2+}$-ATP were obtained by equilibrating 0.4 μL of protein (20.8 mg/mL MtCorBΔC purified in 0.03% DDM; 5 mM ATP added prior to crystallization) and 0.4 μL of reservoir solution (0.1 M MOPS, pH 7.0; 9% PEG 8000; 20 mM MgCl$_2$) in sitting-drop vapor diffusion system incubated at 22 °C. Rod-like crystals appeared after 2 weeks. The crystal was cryo-protected with reservoir solution supplemented with 5 mM ATP and 30% ethylene glycol, picked up in a nylon loop, and flash-cooled in an N$_2$ cold stream.

Crystals of MtCorBΔC$_{\Delta loop}$ with Mg$^{2+}$-ATP were obtained by equilibrating 1 μL of protein (19.5 mg/mL MtCorBΔC$_{\Delta loop}$ purified in 0.05% UDM; 5 mM ATP added prior to crystallization) and 1 μL of reservoir solution (0.1 M sodium citrate, pH 5.5; 0.1 M Li$_2$SO$_4$; 0.1 M NaCl; 20 mM MgCl$_2$; 34% PEG400; 10 mM Na$_2$HPO$_4$) in hanging-drop vapor diffusion system incubated at 22 °C. Petal-like crystals appear after 1 week. The crystal was directly picked up in a nylon loop and flash-cooled in an N$_2$ cold stream.

Crystals of MtCorBΔC R235L were obtained by equilibrating 1 μL of protein (18.4 mg/mL MtCorBΔC R235L purified in HPLC buffer containing 20 mM HEPES, 500 mM NaCl, 0.05% UDM, pH 7.5) and 1 μL of reservoir solution (0.1 M sodium citrate, pH 5.5; 50 mM Li$_2$SO$_4$; 11% PEG4000; 10 mM MgSO$_4$) in hanging-drop vapor diffusion system incubated at 22 °C. Rod-like crystals appear after 1 day. The crystal was cryo-protected with reservoir solution supplemented with 25% glycerol and 0.6 M NaCl, picked up in a nylon loop, and flash-cooled in an N$_2$ cold stream.

**Data collection and structure determination**. The MtCorB$_{CBS}$ dataset from a single crystal was collected using a Pilatus3 6 M detector at beamline 5.0.2 of Advanced Light Source (ALS). Data processing and scaling were performed with HKL-2000[58] with auto-corrections enabled. Initial phases were obtained by molecular replacement with Phaser[59] in PHENIX[60] using the CBS-pair domain structure of EcCorC (PDB: 5YZ2)[61].

The MtCorB$\Delta$C$_{\Delta loop}$ dataset from a single crystal was collected using a Pilatus3 6 M detector at beamline 08ID-1 of the Canadian Macromolecular Crystallography Facility of the Canadian Light Source (CLS). The dataset showed anisotropic diffraction up to 3.25 Å. Data processing and scaling were performed with HKL-2000[58] with auto-corrections enabled, in which ellipsoid truncation was performed automatically. Resolution cut-off is based on CC$_{1/2}$ = 0.3[62]. Resolution limits after ellipsoid truncation were $a^*$ = 4.07 Å, $b^*$ = 3.71 Å, and $c^*$ = 3.25 Å. Initial phases for the CBS-pair domain were obtained by molecular replacement with Phaser[59] in PHENIX[60] using MtCorB$_{CBS}$ structure (determined in this study). Then AutoBuild[33] was used to build in the missing domains (TMD and AHB).

The MtCorB$\Delta$C R235L dataset from a single crystal was collected using Pilatus3 6 M detector at beamline 08B1-1 of the CLS. Data processing and scaling were performed with HKL-2000[58] with auto-corrections enabled. Initial phases were obtained by molecular replacement with Phaser[59] in PHENIX[60] using the CBS-pair and TMD structures (determined in this study).

After molecular replacement, the models were subsequently improved through iterative cycles of manual building with Coot[63] and refinement with phenix. refine[64]. TLS parameters were included at later stages of the refinement[65]. The final structures were validated with MolProbity[66]. Crystallographic data collection and structure refinement statistics are shown in Table 1. Structural images were prepared with PyMOL, Version 2.3.4 (Schrödinger LLC, New York). Electrostatic surface potentials were calculated using the APBS plugin within PyMOL[67].

**Isothermal titration calorimetry**. ITC experiments were performed on a MicroCal VP-ITC titration calorimeter (Malvern Instruments Ltd.) at 20 °C with stirring at 310 rpm. TMD-containing constructs (15 μM final concentration) and ligands were prepared in HPLC buffer containing 0.05% UDM (±50 mM MgCl$_2$). Cytosolic constructs (30 μM final concentration) were prepared in HPLC buffer (±10 or 50 mM MgCl$_2$). The ligands were injected 19 times (5 μL for the first injection, 15 μL for subsequent injections), with 4 min intervals between injections. Results were analyzed using ORIGIN software (MicroCal) and fitted to a binding model with a single set of identical sites.

**Analytical ultracentrifugation**. Sedimentation velocity AUC experiments were performed at 20 °C using a Beckman Coulter XL-I Optima analytical ultra-centrifuge and an An-60Ti rotor at 98,000$g$ (35,000 rpm) for 18 h with scans performed every 60 s. A double-sector cell, equipped with a 12 mm Epon center-piece and sapphire windows, was loaded with 380 and 400 μL of sample and HPLC buffer. MtCorB$_{CBS}$ (100 μM) with various ligands were monitored using inter-ference optics. The data were analyzed with Sedfit v1501b[68] using a continuous c(s) distribution. Numerical values for the solvent density, viscosity, and partial specific volume were determined using Sednterp[69]. Buffer density and viscosity were cal-culated to be 1.0039 g/cm³ and 0.01026 mPa·s, respectively (20 mM HEPES, 100 mM NaCl, pH 7.5). Partial specific volume for MtCorB$_{CBS}$ was calculated to be 0.7464 cm³/g. The frictional ratio ($f/f_0$) value for MtCorB$_{CBS}$ was calculated using US-SOMO[70] to be 1.26. Residual and c(s) distribution graphs were plotted using GUSSI[71].

**Production and purification of MSP1D1**. pMSP1D1 was a gift from S. Sligar (Addgene plasmid 20061). MSP1D1 production was carried out according to published protocols[72]. In brief, *E. coli* BL21 (DE3) cells transformed with pMSP1D1 were grown in LB at 37 °C to an optical density of 0.8 and induced with 1 mM IPTG for 4 h at 30 °C. MSP1D1 was purified by nickel affinity chromato-graphy according to standard conditions described in[72]. The polyhistidine tag was removed by overnight incubation with TEV protease and further purified by Superdex-75 size-exclusion column (GE Healthcare) in HPLC buffer (20 mM HEPES, 150 mM NaCl, pH 7.5).

**Reconstitution into nanodisc**. MtCorB and MSP1D1 were mixed with soybean polar extract (Avanti) solubilized in 40 mM DDM at a MtCorB:MSP1D1:lipid molar ratio of 2:10:550 in HPLC buffer. Detergent was removed by incubation with Bio-Beads (Bio-Rad SM-2 Resin) at 4 °C overnight with constant rotation. Bio-beads were removed via filtration and the reconstitution mixture was re-loaded onto Qiagen Ni-NTA resin to remove empty nanodiscs. The resin was washed with wash buffer (20 mM HEPES, 200 mM NaCl, 20 mM imidazole, pH 7.5) and eluted with wash buffer with 300 mM imidazole. The eluted protein was further purified by size exclusion chromatography on a Superdex 200 Increase 10/300 GL column (GE Healthcare) in HPLC buffer (20 mM HEPES, 150 mM NaCl, pH 7.5).

**Hydrogen deuterium exchange mass spectrometry**. HDX-MS reactions were performed in a similar manner as described previously[73,74]. In brief, HDX reactions for MtCorB and MtCorB$\Delta$C were conducted in a final reaction volume of 10 μL with a molar quantity of 20 pmol of MtCorB and MtCorB$\Delta$C. The reaction was

started by the addition of 9.0 μL of D$_2$O buffer (100 mM NaCl, 20 mM HEPES pH 7.5, 94% D$_2$O (V/V)) to 1.0 μL of protein solution (final D$_2$O concentration of 84.9%). The reaction proceeded for 3, 30, 300, or 3000 s at 20 °C, before being quenched with ice-cold acidic quench buffer, resulting in a final concentration of 0.6 M guanidine–HCl and 0.9% formic acid post quench. All conditions and timepoints were created and run in triplicate. Samples were flash-frozen imme-diately after quenching and stored at −80 °C until injected onto the ultra-performance liquid chromatography (UPLC) system for proteolytic cleavage, peptide separation, and injection onto a QTOF for mass analysis, described below.

Protein samples were rapidly thawed and injected onto a UPLC system kept in a Peltier has driven cold box at 2 °C (LEAP). The protein was run over two immobilized pepsin columns (Trajan; ProDx Protease column PDX.PP01-F32) and the peptides were collected onto a VanGuard Precolumn trap (Waters). The trap was eluted in line with an ACQUITY 1.7 μm particle, 100 × 1 mm² C18 UPLC column (Waters), using a gradient of 5–36% B (Buffer A 0.1% formic acid, Buffer B 100% acetonitrile) over 16 min. MS experiments were performed on an Impact HD QTOF (Bruker) and peptide identification was done by running tandem MS (MS/MS) experiments run in data-dependent acquisition mode. The resulting MS/MS datasets were analyzed using PEAKS7 (PEAKS) and a false discovery rate was set at 1% using a database of purified proteins and known contaminants. HDExaminer Software (Sierra Analytics) was used to automatically calculate the level of deuterium incorporation into each peptide. All peptides were manually inspected for the correct charge state and presence of overlapping peptides. Deuteration levels were calculated using the centroid of the experimental isotope clusters. No correction was made for back exchange during analysis, so all deuterium incorporation values are relative. Differences in exchange in a peptide were considered significant if they met three of the following criteria: >5% change in exchange, >0.4 Da mass difference in exchange, a $p$ value < 0.01 using a two-tailed Student's $t$ test, and change spanned by multiple peptides. The source data are provided as a Source Data file.

**Proteoliposome reconstitution and Mg$^{2+}$ transport assay**. Proteoliposome were made following[75] with modifications. A 3:1 mixture of 1-palmitoyl-2-oleoyl-*sn*-glycero-3-phosphoethanolamine (POPE) and 1-palmitoyl-2-oleoyl-*sn*-glycero-3-phospho-(1'-*rac*-glycerol) (POPG) (Avanti Polar Lipids) were dried into a thin film, followed by overnight incubation in a vacuum chamber. Dried lipids were solubilized to 5 mg/mL in buffer A (20 mM HEPES-KOH, 100 mM KCl, 50 mM NaCl, 0.5 mM EDTA-KOH, pH 7.5) supplemented with 35 mM CHAPS and incubated at room temperature for 2 h. UDM-purified MtCorB and TtCorB or DDM-purified TmCorA were mixed with 100 μL of solubilized lipids (0.5 mg) at a protein/lipid ratio of 1:30, 1:90, 1:270, or 1:810 (wt:wt) and incubated for 20 min at room temperature. For the no-protein liposome control, the same volume of HPLC buffer containing 0.05% UDM was added. Proteoliposomes were formed by adding the protein/lipid sample to a partially dehydrated Sephadex G-50 column (1.5 mL) equilibrated in buffer A. Membrane impermeable magnesium indicator, mag-fura-2 (Thermo Fisher; 1 mM stock in H$_2$O) was added to a final concentration of 50 μM to the proteoliposomes. Dye encapsulation was performed through 1 freeze–thaw cycle (we observed reduced activity with an increased number of cycles) and subsequent extrusion through 0.8 μm polycarbonate filters. Extra-vesicular dye was removed by centrifuging the proteoliposomes through a partially dehydrated Sephadex G-50 column equilibrated in buffer A.

The transport assay was measured with a SpectraMax M5e fluorescence plate reader using SoftMax Pro 5 software. Totally, 10 μL of dye-encapsulated liposomes were diluted with 190 μL of reaction buffer (buffer A without EDTA) in a 96-well black bottom plate (Greiner) and baseline fluorescence was measured for 1 min at 25 °C. The uptake reaction was initiated by the addition of MgCl$_2$, CaCl$_2$, or Zn (CH$_3$CO$_2$)$_2$, and recorded for 10 min. For mag-fura-2-encapsulated liposomes, fluorescence was measured using two excitation wavelengths (330 and 369 nm) and one emission wavelength (509 nm) every 6 s including 1 s shaking between reads. Triton X-100 and EDTA were added to 0.1% and 12.5 mM to obtain the maximum and minimum mag-fura-2 signals for conversion to [Mg$^{2+}$]. All measurements were performed in triplicates. To test for electrogenic transport, the NaCl and KCl in the reaction buffer were replaced with 150 mM NMDG-Cl and 1 μM valinomycin (Sigma; 1 mM stock in DMSO) to generate an outward potassium gradient. For no valinomycin condition, an equal volume of DMSO was added.

**Homology modeling of TtCorB and HsCNNM2**. The homology model of TtCorB and HsCNNM2 were constructed with MODELLER Version 9.25[76] using the MtCorB$\Delta$C structure as the template. The sequence alignment was performed using Clustal Omega[77]. The model was evaluated using PROCHECK[78]. Con-servation coloring was done with ProtSkin (http://www.mcgnmr.mcgill.ca/ProtSkin/).

**Molecular dynamics simulation**. MtCorB$\Delta$C structure was used as the starting model for MD simulations. The transporter was embedded in a bilayer of 3POPC:1POPG lipids and solvated in 150 mM NaCl (neutral with 277 Na$^+$ and 139 Cl$^−$ ions) using the web service CHARMM-GUI[79,80]. Most residues were assigned their standard protonation state at pH 7. The total number of atoms in the atomic model is on the order of 200,000. The all-atom CHARMM force field

PARAM36 for protein[81–83], lipids[84], and ions[85] was used. Explicit water was represented with the TIP3P model[86]. The model was refined using energy minimization for at least 2000 steps. All the simulations were performed under NPT (constant number of particle N, pressure P, and temperature T) conditions at 303 K and 1 atm, and periodic boundary conditions with electrostatic interactions were treated by the particle mesh Ewald method[87] and a real-space cutoff of 12 Å. The simulations use a time step of 2 fs, with bond distances involving hydrogen atoms fixed using the SHAKE algorithm[88]. After minimization, positional restraints on all of the Cα atoms were gradually released after which a trajectory of 500 ns was generated using NAMD version 2.11[89]. The same process was applied to the model of symmetric MtCorBΔC dimer, in which the simulation ran for 124 ns.

**Targeted molecular dynamics simulation**. For preparing the target (open) model, an $Mg^{2+}$ ion was pulled through the interface helices of MtCorBΔC while keeping a distance restraint with Glu111 of both protomers. This caused the two helices to open. All forces were then removed and the whole system was equilibrated for ~8 ns. For the targeted MD simulation, the MtCorBΔC structure was used as the starting model. During the simulation, forces were applied on the backbone of the transmembrane helices with a force constant of 200 kcal mol$^{-1}$ A$^{-2}$ for a period of 10 ns. This decreases the RMSD value linearly to reach the final target conformation.

**Reporting summary**. Further information on research design is available in the Nature Research Reporting Summary linked to this article.

## Data availability

Data supporting the findings of this manuscript are available from the corresponding author upon reasonable request. A reporting summary for this article is available as a Supplementary Information file. Atomic coordinates and structure factors for MtCorB$_{CBS}$ with $Mg^{2+}$-ATP, MtCorBΔC with $Mg^{2+}$-ATP, and MtCorBΔC R235L mutant have been deposited to the Protein Data Bank (www.rcsb.org) under accession numbers 7MSU, 7M1T, and 7M1U, respectively. Source data are provided with this paper.

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

## Acknowledgements

We thank Katharina Dürr and her lab members for assistance with the initial screening of CorB orthologs; Alexei Gorelik for advice on crystallographic data processing; Nazzareno D'Avanzo for assistance with liposome transport assay; Satchal Erramilli for the TmCorA plasmid and nanodisc protocol; Martin Schmeing and Camille Fortinez for crystallographic data collection at ALS; Canadian Light Source (CLS) staff: Michel Fodje, Shaun Labiuk, Bulat Gabidullin, and Kathryn Janzen. CLS is supported by the Canada Foundation for Innovation, the Natural Sciences and Engineering Research Council (NSERC), the National Research Council, the Canadian Institutes of Health Research (CIHR), the Government of Saskatchewan, and the University of Saskatchewan. Y.S.C. is supported by a Canadian Institutes of Health Research Doctoral Research Award (GSD-167011). This work is supported by Natural Sciences and Engineering Research Council of Canada grant RGPIN-2020-07195 to K.G.

## Author contributions

Y.S.C. designed experiments, cloned constructs, performed small-scale screenings, purified proteins, solved crystal structures, and performed liposome transport assays. G.K. assisted with crystal screening and performed ITC experiments. B.E.M. performed HDX-MS experiments. A.R. performed molecular dynamics simulations. R.F. performed AUC and ITC experiments of soluble domains. Y.S.C. and K.G. wrote the paper. K.G., J.E.B., and B.R. oversaw the research.

## Competing interests

The authors declare no competing interests.
