## [Peer Review File · Nature Communications]

Reviewers' comments:

Reviewer #1 (Remarks to the Author):

CNNM are a ubiquitous but largely enigmatic class of transmembrane proteins found in all domains of life that have been implicated primarily in Mg^{2+} transport and associated with rare human disease. In the current work, Chen et al. report the crystal structure of an ortholog of the CNNM protein from *Methanococcus thermophilus*, MtCNNM, which was truncated at the C-terminal CorC domain to facilitate crystallization. Overall, this work adds important new insights to an emerging body of biochemical, biophysical and structural studies on the CNNM proteins from this group and others including the recent determination of the full-length homologous CorC protein reported by Huang et al. [Sci Adv, 2021; PMID: 28246390], but important clarifications, additional analyses, and key control experiments are needed before this work should be considered for publication. The general presentation of the manuscript (i.e. the introduction and results section) should also be considerably revised in order for naïve readers to better appreciate the context and interpretation of the system and results presented.

Essential revisions:

1) Given the recent and thorough publication by Huang et al., the authors need to appreciate that their report is not the first description of the DUF21 domain fold, and should remove the terms 'novel' or 'new' throughout their descriptions. In fact, a systematic comparison of their structure and analysis in light of the results presented by Huang et al. would provide a more robust and informative framework to advance our knowledge of this understudied transport protein family. For example, the TMD structure from Huang et al. is a symmetric homodimer, and superposition analysis with MtCNNM may provide informative insights. More importantly, Huang et al. provide strong evidence for a TMD-bound Mg^{2+} coordination site and further implicate a Na^{+} -binding site – it is therefore instructive for the current authors to provide a comparative analysis of their structure and conclusions to these recently published results (e.g. it is unclear if the Na^{+} the present authors assign occurs in the same site assigned as a Mg^{2+} by Huang et al.; also see point below on expected coordination geometry for Mg^{2+} in general)

2) The extent of the "functional data" data presented in the current study include a cellular reconstitution assay and not a direct electrophysiological or transport assay; therefore, the authors are over stating their "proof" that their "structure clearly demonstrates CNNMs are ion transporters rather than regulators of other proteins." IMO, they have not formally provided this proof, nor formally ruled out other (indirect or regulatory) possibilities, and the limitations of their methods should be appropriately stated. Similarly, the authors have not provided formal proof that their structures are "of a CNNM protein in both an active and inactive conformation", this is simply their speculation. At least hydration analysis by molecular dynamics studies should be considered to potentially substantiate these types of speculations. Additionally, about general presentation, the authors do not clearly provide statements about the "inward facing" state of their TMD containing structures, and this important concept and result needs to be introduced and stated early on.

3) The authors should realize *Methanococcus thermophilus* is in the Domain Archaea (Kingdom Euryarchaeota) and so should correct the title of their manuscript for accuracy because the current form indicates the structure of a bacterial protein is being presented.

4) It would be helpful to learn the conditions under which the protein has been crystallized within the results section, i.e. were any divalent cations (or metal chelators) added during purification or crystallization, other salt concentrations, detergent based matrix, etc, as well as the available resolution of the diffraction data. This would help the reader begin to understand if this may represent a truly apo-conformation of the protein, and why Mg^{2+} was not also considered to be present in the presented structure.

5) For the mutational analysis, please indicate which each residue has been mutated to (i.e. is the S21 mutant studied in MpfA really the equivalent of S269W in human CNNM2), and compare the present results to the extensive mutational analyses presented by Huang et al. (i.e. agreement, divergence or added insights). More importantly, the authors minimally have to demonstrate the expression level of their mutant proteins are expressed at near wild-type levels for all mutants studied, otherwise it is impossible to understand the relevance of the suggested phenotypes (i.e. for all constructs reported in Sup Fig 5).

6) It is unclear why the authors commit a section of the manuscript called "Lipid binding interface". The assignment of detergent molecules and sulfate, present at high concentrations in the crystallization buffer, does not constitute evidence for any physiological relevance, nor do the authors attempt to substantiate this claim. The authors should please reconsider how they intend to present this description.

7) Putative Mg²⁺ coordinated to the AHB – as per point #2 above, is the first the possibility that Mg²⁺ might be present in the crystallization condition is only introduced here by the authors. More importantly, the Mg²⁺ – O distances indicated by the authors are not at all consistent with the expected strict hexagonal coordination shell and distances that typically define Mg²⁺ coronation (e.g. distances in the 2.1-2.2 Å range, see references below), and so the likelihood that this density can be confidently assigned is highly questionable.

References:

I. Persson, Hydrated metal ions in aqueous solution: How regular are their structures? Pure Appl. Chem. 82, 1901–1917 (2010).

Harding MM (2004) The architecture of metal coordination groups in proteins. Acta Crystallogr D Biol Crystallogr 60: 849–859.

Maguire ME, Cowan JA (2002) Magnesium chemistry and biochemistry. Biometals 15: 203–210

8) The molecular dynamics studies and their potential implications should be better described, in particular the steered Na⁺ simulations. In general, throughout the manuscript, the authors fail to describe the MtCNNM structure (TMD) as inward facing, occluded, or outward facing, an important oversight that needs to be reconsidered in the revised manuscript in order for the reader to better appreciate what functional state(s) of the transporter might be being described.

9) Overall, the authors appear to describe the MtCNNM and CNNM proteins in general as transporters, yet refer to the TMD as having a "channel cavity". The functional differences between channels and transporters, and the gating cycles and structural features for each should be better appreciated or even defined by here by the authors to clarify their intended description.

10) Given the limited resolution and high anisotropy of the data set for the deltaCorC structure, it is inappropriate for the authors to present 2Fo-Fc maps as evidence of the quality of the map or model. Please replace these all instead with calculated SA-omit maps.

Reviewer #2 (Remarks to the Author):

In this manuscript by Chen et al., a structural analysis of archaean CNNM homologue magnesium transporter is presented. The authors describe the structures of MtCNNM with the deletion of CorC domain and a loop; a structure of isolated CBS dimer in the presence of Mg²⁺ ATP, and the structure of a mutant unable to bind ATP.

They also studied the contributions of cation and ATP binding residues via mutagenesis and

complementation studies, performed ITC, AUC and deuterium exchange experiments to thoroughly characterise ATP binding to CBS. In addition MD simulations were performed to study the conformational changes during the transport. Furthermore, the authors used the obtained structures to fabricate a model of human CNNM to get an insight into the effect of disease-linked mutations on the structure.

Despite it is a high-quality study and conclusions are supported by the obtained results, this work, and I understand this is sad, but unfortunately true, is not novel. Very similar results have been reported recently by another group, see *Sci Adv* 7 (7), eabe6140. DOI: 10.1126/sciadv.abe6140. Therefore I cannot recommend this manuscript for publication in *Nature Communications*. The authors need completely rewrite their manuscript and focus on a comparison with the published structure of CorC protein.

Some comments which might help during the revision - the reported sodium site seems to be assigned to a magnesium-binding site (*Sci Adv* 7 (7), eabe6140), double check this by refinement of Mg²⁺ instead of Na⁺ at this site. The distance-wise it is close to dehydrated Mg²⁺ ion. In the methods section, the reconstitution into nanodiscs is mentioned, however I do not find any reference to this in the main text. In general almost all figures are overcrowded with too many panels, try to keep only the essential ones and put the rest into the supplementary.

Reviewer #3 (Remarks to the Author):

Manuscript#: NCOMMS-21-05862

The manuscript presented by Chen et al. is a very interesting and detailed study that describes the crystal structure of a bacterial protein MtCNNM that shares (only partially) the domain distribution of the eukaryotic CNNM family of magnesium homeostatic factors. Two different conformational states have been isolated in the absence and in the presence of MgATP that serve to hypothesize a possible mechanism of transport. Beyond that, the most remarkable finding of this work is the unveiling of the N-terminal region of the protein which consists of a DUF21 transmembrane domain, whose fold remained unknown so far. The actual function of DUF21 itself and the proteins carrying this module has been under debate in the last years. Two main functions, either as magnesium sensors or as direct transporters have been proposed for CNNM-like proteins. The structures of the DUF21 domain by Gehring's team provide a relevant piece of the puzzle and support its involvement in direct cation transport. Overall, the study is technically sound, the manuscript is clearly written and understandable to a wide audience. Nevertheless, in my opinion, the manuscript as it stands tries to draw some conclusions that are not sufficiently sustained by experimental data. Accordingly, several major issues should be addressed prior to publication.

My first concern is related to the direct attribution of MtCNNM as a bacterial homolog of a CNNM transporter. MtCNNM shows a substantial number of differences from known CNNMs, which invite caution and do not allow it to state so emphatically that it performs the same function. MtCNNM shows a poor sequence identity with all known CNNMs. Besides that, it contains a CorC domain and not a CNBH module at the C-terminal region. This is relevant, as the function, the location, the interaction with other protein partners, and the structural effect of the CorC domain may differ significantly from those played by a CNBHD. In addition, MtCNNM lacks other regions of CNNMs, as the unstructured C-tail following the CNBH domain or the N-terminal extracellular region preceding the DUF21 domain. Additionally, MtCNNM affinity for MgATP at the CBS domains is significantly lower than in CNNMs, which might also be indicative of a distinct function. For this reason, and until more solid evidence is obtained, MtCNNM should not be designed as a bacterial CNNM. This needs to be made clear both, in the title and along with the text.

Nevertheless, my major concerns refer to concrete aspects of the structural interpretation that directly affect the transport mechanism and the transitions from the apo- to the MgATP-bound state as detailed below:

(1) Asymmetry of the MtCNNM dimer.

In the MgATP-bound MtCNNM crystals, the protein self-associate in non-symmetric dimers. In contrast, the nude MtCNNM dimers appear to maintain pseudo-2-fold symmetry. This point is key, as the cytoplasmic and intermembrane domains when analyzed pair-by-pair with their complementary counterparts in both types of conformers, appear to maintain local 2-fold symmetry. Previous studies by Gehring's team and other laboratories also indicate that CNNMs dimers likely form 2-fold symmetric assemblies. For example, crystals of the isolated CBS-domain pair (Bateman module) from CNNM2, CNNM3, and CNNM4, no matter whether they were grown in the absence or in the presence of MgATP and/or PRL phosphatases, clearly form 2-fold symmetric assemblies. Pointing in the same direction, isolated CNBH domains from CNNM2, CNNM3, and CNNM4 associate in 2-fold dimers. Similarly, the DUF21 domains in MtCNNM are related by a 2-fold axis, both in the absence and in the presence of MgATP. A 180 degrees rotation also relates the Bateman modules in MgATP-bound MtCNNM. Moreover, solution studies on the cytoplasmic region of CNNM4 (Giménez-Mascarell, 2019) by SAXS recently showed 2-fold symmetric assemblies. Taken together, all these data invite us to think that the lack of symmetry observed in the MgATP-MtCNNM dimers might derive from the interactions with neighboring molecules in the crystals and facilitated by the flexibility and length of the linkers connecting the different protein domains (for example the AHB helices with the DUF21 domain, or the latter with the DUF21 domain). In fact, the model proposed in Figure 5e reflects a transition based on symmetric dimers.

A similar disruption of the 2-fold symmetry, permitted by the flexibility and length of the linker connecting the CBS-pair domain with the CNBH domain, could also explain the asymmetry formerly observed in crystals of the cytoplasmic domain of CNNM2 (PDB entry 6N7E, Chen et al, Structure, 28, 324-335, 2020).

Thus, it is important to take into account that the asymmetry observed in the crystals could be artifactual and not maintained in solution, where no neighboring molecules can distort the dimer. A comparative MD analysis considering symmetric dimers (like for example the one depicted in the attached pictures) should be performed and discussed in the manuscript. For example, the authors should explore by MD whether the orientation of the AHB helix of molecule B containing residues (166-181) gets anomalously trapped by helix 35-46 from the DUF21 domain, thus deriving the dimer towards an asymmetric assembly. The transition from the apo- to the ATP-bound form should also be revisited and compared with the asymmetric option.

(2) Effect of mutation R235L: Interestingly, mutation R235L impairs ATP binding at the CBS domains but does not appear to cause further effects in the Bateman module of MtCNNM. Under this scenario, how do the authors explain the dramatic reorientation of the Bateman modules and the drastic conformational change experienced by the entire dimer? It is well documented (for example visit articles describing the effect of nucleotides in IMPDH, CBS, or CNNMs) that binding of nucleotides at Bateman modules disrupts the interactions previously existing between the two CBS motifs. These changes are known to determine their relative orientation and their contact with the rest of the protein. This aspect should be discussed in the revised version of the manuscript.

(3) A movie showing the transformation from the unbound to the MgATP dimer should be provided as Supplemental Material. This would help readers understand the transitions experienced by the protein upon binding ATP.

(4) Minor issue: The authors have placed a glycerol molecule in the model refined from the low-resolution C2 crystals. In fact, the superimposition of the two protein monomers suggests that the electron density has been wrongly attributed to a glycerol molecule, which coincides with part of the protein linker in the other monomer. Thus, this glycerol is very likely not present. Please correct this error in the PDB file, refine the structure accordingly and validate the model again.

Reviewers' comments:

Reviewer #1 (Remarks to the Author):

CNNM are a ubiquitous but largely enigmatic class of transmembrane proteins found in all domains of life that have been implicated primarily in Mg²⁺ transport and associated with rare human disease. In the current work, Chen et al. report the crystal structure of an ortholog of the CNNM protein from *Methanococcus thermophilus*, MtCNNM, which was truncated at the C-terminal CorC domain to facilitate crystallization. Overall, this work adds important new insights to an emerging body of biochemical, biophysical and structural studies on the CNNM proteins from this group and others including the recent determination of the full-length homologous CorC protein reported by Huang et al. [Sci Adv, 2021; PMID: 28246390], but important clarifications, additional analyses, and key control experiments are needed before this work should be considered for publication.

We thank the reviewer for the critical comments and the opportunity to improve the manuscript. We have extensively rewritten the paper to cite the study by Huang et al. [Sci Adv, 2021; PMID: 28246390] and added new experiments that demonstrate that archaeal and bacterial CNNMs transport Mg²⁺ ions.

We feel it is important to point out that Huang et al. did not determine the structure of the full-length CorC protein. They crystallized the transmembrane and CBS-pair domains separately (without the C-terminal CorC domain) and then pieced the two domains together, generating a theoretical model. We report complete structures of the transmembrane and soluble CBS-pair domains in two different conformations. Our structures allow visualization of the interdomain linkers and contacts responsible for regulating transporter function.

We have also added functional assays in a new Figure 6 that show Mg²⁺ transport activity in liposomes with purified archaeal and bacterial CNNM proteins. These are the first experiments that directly implicate CNNM proteins in Mg²⁺ transport and resolve a major open question about their biochemical function.

The general presentation of the manuscript (i.e. the introduction and results section) should also be considerably revised in order for naïve readers to better appreciate the context and interpretation of the system and results presented.

The introduction and results section have been considerably revised to include more details for naïve readers to better appreciate the context and interpretation of the system and results presented.

Essential revisions:

1) Given the recent and thorough publication by Huang et al., the authors need to appreciate that their report is not the first description of the DUF21 domain fold, and should remove the terms 'novel' or 'new' throughout their descriptions.

We have removed the term "novel" in describing our DUF21 domain fold. Our initial manuscript was prepared prior to publication of the Huang paper and submitted the day after its publication. We have revised our manuscript comprehensively and fully acknowledge the results of Huang et al.

In fact, a systematic comparison of their structure and analysis in light of the results presented by Huang et al. would provide a more robust and informative framework to advance our knowledge of this understudied transport protein family. For example, the TMD structure from Huang et al. is a symmetric homodimer, and superposition analysis with MtCNNM may provide informative insights.

We have now added systematic comparisons of our results to that of Huang et al in the results section.

More importantly, Huang et al. provide strong evidence for a TMD-bound Mg²⁺ coordination site and further implicate a Na⁺-binding site – it is therefore instructive for the current authors to provide a comparative analyses of their structure and conclusions to these recently published results (e.g. it is unclear if the Na⁺ the present authors assign occurs in the same site assigned as a Mg²⁺ by Huang et al.; also see point below on expected coordination geometry for Mg²⁺ in general)

Due to the limited resolution of our structures, we cannot measure the distance and geometry around the bound ion precisely enough to definitely assign it as Na⁺ or Mg²⁺. For reasons given in the first submission, we chose to model the density as a Na⁺ ion; however, given the higher resolution of the Huang structure, we have revised our structure to include a Mg²⁺ ion.

2) The extent of the “functional data” data presented in the current study include a cellular reconstitution assay and not a direct electrophysiological or transport assay; therefore, the authors are overstating their “proof” that their “structure clearly demonstrates CNNMs are ion transporters rather than regulators of other proteins.” IMO, they have not formally provided this proof, nor formally ruled out other (indirect or regulatory) possibilities, and the limitations of their methods should be appropriately stated.

We agree with the reviewer and apologize for the overstatement. While suggestive, the structure and experiments in cells do not prove that CNNMs are ion transporters. To address that, we have added liposome transport assays that demonstrate direct Mg²⁺ transport by the CorB/CNNM proteins (Figure 6). These studies are the first measurements of transport activity with purified CNNM proteins and definitively rule out a dependence on other proteins for transport activity.

Similarly, the authors have not provided formal proof that their structures are “of a CNNM protein in both an active and inactive conformation”, this is simply their speculation. At least hydration analysis by molecular dynamics studies should be considered to potentially substantiate these types of speculations.

We apologize for the overstatement. We have now re-phrased it with more speculative wording as part of a proposed inactivation mechanism. To substantiate the inward-facing conformation, we have added MD hydration analysis of our crystal structure (Supplementary Fig. 10a).

Additionally, about general presentation, the authors do not clearly provide statements about the “inward facing” state of their TMD containing structures, and this important concept and result needs to be introduced and stated early on.

We have now provided statements about the “inward-facing” state of our TMD structure early on in the results section. We regret the imprecise language of the initial submission.

3) The authors should realize *Methanococcus thermophilus* is in the Domain Archaea (Kingdom Euryarchaeota) and so should correct the title of their manuscript for accuracy because the current form indicates the structure of a bacterial protein is being presented.

We apologize for the confusion. The original title should have said prokaryotic. The title has been worded to clarify that the crystal structures are of an archaeal protein.

4) It would be helpful to learn the conditions under which the protein has been crystallized within the results section, i.e. were any divalent cations (or metal chelators) added during purification or

crystallization, other salt concentrations, detergent based matrix, etc, as well as the available resolution of the diffraction data. This would help the reader begin to understand if this may represent a truly apo-conformation of the protein, and why Mg²⁺ was not also considered to be present in the presented structure.

The results sections have been considerably revised to include details about the protein crystallization conditions as well as available resolution of diffraction data.

5) For the mutational analysis, please indicate which each residue has been mutated to (i.e. is the S21 mutant studied in MpfA really the equivalent of S269W in human CNNM2), and compare the present results to the extensive mutational analyses presented by Huang et al. (i.e. agreement, divergence or added insights). More importantly, the authors minimally have to demonstrate the expression level of their mutant proteins are expressed at near wild-type levels for all mutants studied, otherwise it is impossible to understand the relevance of the suggested phenotypes (i.e. for all constructs reported in Sup Fig 5).

Due to Covid-19 restrictions and the subsequent closure of our collaborator's laboratory in Geneva, we are unable to perform the experiments to confirm the expression levels of the MpfA mutants. The preparation and analysis of *S. aureus* cell extracts requires specialized facilities and training that are not available to us at present. While we retain confidence in the conclusions of the MpfA mutational analysis, we have decided to drop those results in favor of liposome-based Mg²⁺ transport assays that directly show transport activity by CorB proteins.

Figure 6 now presents a set of Mg²⁺ transport assays in vesicles with both the archaeal MtCorB and a eubacterial ortholog from the gram-negative bacterium *Tepidiphilus thermophilus*. While qualitatively similar, we observed higher transport rates with the *T. thermophilus* protein, which allowed more quantitative analysis of mutants designed based on our crystal structures. The Mg²⁺ transport assays are accompanied by gel-filtration profiles of the mutant proteins as well as SDS-PAGE analysis of the resulting proteoliposomes to confirm reconstitution. In the same section, we have also included comparison of the mutant results to that of Huang et al.

6) It is unclear why the authors commit a section of the manuscript called “Lipid binding interface”. The assignment of detergent molecules and sulfate, present at high concentrations in the crystallization buffer, does not constitute evidence for any physiological relevance, nor do the authors attempt to substantiate this claim. The authors should please reconsider how they intend to present this description.

We thank the reviewer for the suggestion and have re-named the section heading to “A re-entrant juxtamembrane helix encircles TM helices”.

7) Putative Mg²⁺ coordinated to the AHB – as per point #2 above, is the first the possibility that Mg²⁺ might be present in the crystallization condition is only introduced here by the authors. More importantly, the Mg²⁺ – O distances indicated by the authors are not at all consistent with the expected strict hexagonal coordination shell and distances that typically define Mg²⁺ coronation (e.g. distances in the 2.1-2.2 Å range, see references below), and so the likelihood that this density can be confidently assigned is highly questionable. References: I. Persson, Hydrated metal ions in aqueous solution: How regular are their structures? Pure Appl. Chem. 82, 1901–1917 (2010). Harding MM (2004) The architecture of metal coordination groups in proteins. Acta Crystallogr D Biol Crystallogr 60: 849–859. Maguire ME, Cowan JA (2002) Magnesium chemistry and biochemistry. Biometals 15: 203–210

The Mg²⁺ near the AHB has now been removed from the model.

8) The molecular dynamics studies and their potential implications should be better described, in particular the steered Na⁺ simulations.

We have included more descriptions in the MD section. We also added two supplemental figures (Supplementary Fig. 10) showing time-dependent variations of the targeted MD simulation.

In general, throughout the manuscript, the authors fail to describe the MtCNNM structure (TMD) as inward facing, occluded, or outward facing, an important oversight that needs to be reconsidered in the revised manuscript in order for the reader to better appreciate what functional state(s) of the transporter might be being described.

We have now provided statements about the “inward-facing” state of our TMD structure early on in the results section. The statements about the possibility of having additional states (i.e. outward-facing) have now been added in the discussion section.

9) Overall, the authors appear to describe the MtCNNM and CNNM proteins in general as transporter, yet refer to the TMD as having a “channel cavity”. The functional differences between channels and transporters, and the gating cycles and structural features for each should be better appreciated or even defined by here by the authors to clarify their intended description.

We apologize for the confusion. We have now removed channel-related terms in our descriptions.

10) Given the limited resolution and highly anisotropy of the data set for the deltaCorC structure, it is inappropriate for the authors to present 2Fo-Fc maps as evidence of the quality of the map or model. Please replace these all instead with calculated SA-omit maps.

We thank the reviewer for the helpful suggestion. The 2Fo-Fc maps in Fig. 3 and S3a have now been changed to show simulated annealing 2Fo-Fc composite omit maps.

Reviewer #2 (Remarks to the Author):

In this manuscript by Chen et al., a structural analysis of archaean CNNM homologue magnesium transporter is presented. The authors describe the structures of MtCNNM with the deletion of CorC domain and a loop; a structure of isolated CBS dimer in the presence of Mg²⁺ ATP, and the structure of a mutant unable to bind ATP. They also studied the contributions of cation and ATP binding residues via mutagenesis and complementation studies, performed ITC, AUC and deuterium exchange experiments to thoroughly characterise ATP binding to CBS. In addition MD simulations were performed to study the conformational changes during the transport. Furthermore, the authors used the obtained structures to fabricate a model of human CNNM to get an insight into the effect of disease-linked mutations on the structure.

Despite it is a high-quality study and conclusions are supported by the obtained results, this work, and I understand this is sad, but unfortunately true, is not novel. Very similar results have been reported recently by another group, see Sci Adv 7 (7), eabe6140. DOI: 10.1126/sciadv.abe6140. Therefore I cannot recommend this manuscript for publication in Nature Communications.

We thank the reviewer for the positive comments. However, we respectfully disagree with the statements that “this study is not novel” and “very similar results have been reported”. While a highly significant result, the novelty of the Science Advances paper by Huang et al. is limited to the

transmembrane domain. For the reasons below, we strongly believe our results, independently obtained, are distinct and worthy of publication. Our results are complementary and extend those of Huang et al. in multiple aspects.

Firstly, our structure is the first crystal structure of CNNM from an archaeon, whereas that determined by Huang et al. is from a gram-negative bacterium. CNNM proteins are present in all eukaryotes, archaea and most bacteria. There are 50,000 protein sequences known; the transmembrane domain of CNNM proteins constitutes the largest family of protein domains of unknown function. The archaeal and bacterial proteins share only 37% sequence identity with distinct differences in the cytosol facing helices of the transmembrane domain (Supplementary Fig. 4d).

The relevance of CNNM proteins in human diseases makes modeling the structure of the human proteins of particular interest. The archaeal and bacterial proteins are only slightly more similar to each other than to the human protein. Structures of both allows better modelling of the human proteins particularly in regions such as the acidic helical bundle or first CBS motif that are more similar in sequence to the archaeal proteins. An alignment of the three protein sequences (human CNNM2, archaeal and bacterial) is shown here.

```

Human/1-875      241 TKMIVGEEK- KFL PFWLQVIFISLLCLSGMFSGLNLGLMALDPMELRIVQNCGT EKEK 299
Archaea/1-426    1  - - - - - MVVIDLLIVE- VVLFIAALLFSGFFSSSEVALISITRAKVHALQSQGRKGAK 51
Bacteria/1-445   1  - - MTPMDD SMLADL- LWQ- WIGLAVLLLTSGFASMSSETVMMANRYRLKARADNGERGAQ 56

Human/1-875      300 NYAKRIEPVRRQGNYLCSLLGNVLVNTTTLTILLDD- - - - - IAGSGLVAVVSTIG 351
Archaea/1-426    52  - - - - ALDTLKRSTDAIQITTLIGSTIANVAVASLATAIGITLYGNL- - GI AVGLVVA AVL 105
Bacteria/1-445   57  - - - - LAAALTAHPERMSVILLVNNAVNVGAATLASVITIELFGQNETMLAVGSFVLTFL 112

Human/1-875      352 I V I F G E I V P Q A I C S R H G L A V G A N T I F L T K F F M M M T F P A S Y P V S K L L D C V L - - - - - 401
Archaea/1-426    106 V L V F G E I G P K M Y A S R Y T E E L A L R V S R P I L F F S K L L Y P - - - - - V L W V T D R I E Q Q F 154
Bacteria/1-445   113 I L V F S E I T P K V I G A R Y A D L L A P Y I A Y P L T A I L R L V G P V V D F V N L F V K G L L W L L R L P R R A T 172

Human/1-875      402 G Q E - - - I G T V Y N R E K L L E M L R V T D P Y N D L V K E E L N I I Q G A L E L R T K T V E D V M T P L R D C F M 458
Archaea/1-426    155 A F R P G V T E P V V T E E E I K E W I D V G E E E G T I E E E E R D M L Y S V L R F G D T T A R E V M T P R V D V M 214
Bacteria/1-445   173 P Q A P S L - - - - - E E L R S L V - - L E S R V L R S E K H R D V L L K L F D L E R I T V A D V M I P R Q A I E F 223

Human/1-875      459 I T G E A I L D F N T M S E I M E S G Y T R I P V F E G E R S N I V D L L F V K D L A F V D P D D C T P L K T I T K F Y 518
Archaea/1-426    215 I E D T A T L E - S A L A I F N E T G F S R I P V Y H E R I D N I V G L L N V K D V F S A V F R Q Q - T S A T I R D L M 272
Bacteria/1-445   224 L D L T D D E E - T L R A Q L A T A Y H T R L P V I E G N P D E V L G I L H V R Q L L A E T L T S G F S R E A I R R S L 282

Human/1-875      519 N H P L H F V F N D T K L D A M L E E F K K G K S H L A I V Q R V N N E G E G D P F Y E V L G I V T L E D V I E E I I K 578
Archaea/1-426    273 - Y E P Y F I P E S K K I D E L L K E L Q V K K Q H M A V V L - - - - - D E Y G S F A G I V T V E D M L E E L V G 323
Bacteria/1-445   283 - S P P Y F V P E E T N A M T Q L Q F Q E H H Q R L A L V V - - - - - D E Y G E L Q G L V T L D D I I E E M V G 333

```

	Human	Archaea	Bacteria
Human	100%	26%	29%
Archaea		100%	37%
Bacteria			100%

Secondly, our structures are more complete and contain both the transmembrane and soluble domains. Huang et al. crystallized the transmembrane and soluble domains separately, which were then pieced together, generating a theoretical model with speculative connections between the transmembrane and soluble domains. With a more complete construct, we were able to image the linkers connecting the domains in two different conformations.

Thirdly, we determined structures of two distinct conformations (apo and Mg²⁺-ATP bound). The apo structure (Fig. 5) shows major changes in domain organization and is supported by HDX-MS and mutagenesis results. The structure rationalizes the disease-associated R407L mutation in

human CNNM4 and allows us to propose a mechanism for transporter regulation by Mg^{2+} -ATP binding. We recently improved the resolution of the apo structure and can now describe in more detail the novel interdomain contacts.

Finally, since submission, we have completed a series of liposome-based transport assays that unambiguously demonstrate Mg^{2+} transport by CNNM/CorB proteins (Fig. 6). These results answer the most important question in the field: whether CNNMs are themselves Mg^{2+} transporters or rather regulators of other Mg^{2+} transporters. The experiments reported by Huang et al. used transport assays in HEK293 cells, which precludes the differentiation of a direct or indirect role in transport. Our new data are the first report of assays with purified CNNM proteins.

The authors need completely rewrite their manuscript and focus on a comparison with the published structure of CorC protein.

We apologize for failing to properly cite the work of Huang et al. Our initial manuscript was written and ready for submission before publication that study. In hindsight, it might have been better to wait and rewrite our manuscript as suggested. We have now added systematic comparisons of our results to those of Huang et al. in our results section.

Some comments which might help during the revision - the reported sodium site seems to be assigned to a magnesium-binding site (Sci Adv 7 (7), eabe6140), double check this by refinement of Mg^{2+} instead of Na^+ at this site. The distance-wise it is close to dehydrated Mg^{2+} ion.

Due to the limited resolution of our structures, we cannot confidently determine the identity of the ion. Based on the structure from Huang et al., we have now placed a Mg^{2+} ion at the site.

In the methods section, the reconstitution into nanodiscs is mentioned, however I do not find any reference to this in the main text.

The reference to nanodisc reconstitution was mentioned together with the HDX-MS results on page 10, line 5: “MtCNNM and MtCNNM Δ C were reconstituted into nanodiscs and hydrogen exchange monitored in the presence and absence of Mg^{2+} -ATP (Supplementary Fig. 8a).” This sentence has been expanded to highlight the nanodisc reconstitution.

In general almost all figures are overcrowded with too many panels, try to keep only the essential ones and put the rest into the supplementary.

We apologize for the overcrowded figures. We have condensed Fig. 2, and Fig. 3 has now been split into two figures (Fig. 3 & 4).

Reviewer #3 (Remarks to the Author):

Manuscript#: NCOMMS-21-05862

The manuscript presented by Chen et al. is a very interesting and detailed study that describes the crystal structure of a bacterial protein MtCNNM that shares (only partially) the domain distribution of the eukaryotic CNNM family of magnesium homeostatic factors. Two different conformational states have been isolated in the absence and in the presence of MgATP that serve to hypothesize a possible mechanism of transport. Beyond that, the most remarkable finding of this work is the unveiling of the N-terminal region of the protein which consists of a DUF21 transmembrane domain, whose fold remained

unknown so far. The actual function of DUF21 itself and the proteins carrying this module has been under debate in the last years. Two main functions, either as magnesium sensors or as direct transporters have been proposed for CNNM-like proteins. The structures of the DUF21 domain by Gehring's team provide a relevant piece of the puzzle and support its involvement in direct cation transport. Overall, the study is technically sound, the manuscript is clearly written and understandable to a wide audience. Nevertheless, in my opinion, the manuscript as it stands tries to draw some conclusions that are not sufficiently sustained by experimental data. Accordingly, several major issues should be addressed prior to publication.

We thank the reviewer for the positive comments and hope our revised manuscript satisfactorily addresses the nomenclature and asymmetry issues.

My first concern is related to the direct attribution of MtCNNM as a bacterial homolog of a CNNM transporter. MtCNNM shows a substantial number of differences from known CNNMs, which invite caution and do not allow it to state so emphatically that it performs the same function. MtCNNM shows a poor sequence identity with all known CNNMs. Besides that, it contains a CorC domain and not a CNBH module at the C-terminal region. This is relevant, as the function, the location, the interaction with other protein partners, and the structural effect of the CorC domain may differ significantly from those played by a CNBHD. In addition, MtCNNM lacks other regions of CNNMs, as the unstructured C-tail following the CNBH domain or the N-terminal extracellular region preceding the DUF21 domain. Additionally, MtCNNM affinity for MgATP at the CBS domains is significantly lower than in CNNMs, which might also be indicative of a distinct function. For this reason, and until more solid evidence is obtained, MtCNNM should not be designed as a bacterial CNNM. This needs to be made clear both, in the title and along with the text.

While we feel that using the same name for the proteins from eubacteria, archaea, and eukaryotes would be preferable, we respect the reviewer's opinion and have changed the name to CorB, which was the first genetic locus identified for a CNNM-like protein (Gibson et al. 1981; PMID: 1779764). Huang et al. chose to call the protein they studied CorC. The *corC* gene was identified in the same paper as *corB* but encodes a protein without the transmembrane DUF21 domain (UniProt: P0A2L3). Without belaboring a point that we have already conceded, it is worth mentioning the strong sequence conservation between the prokaryotic and eukaryotic proteins is suggestive of a common function. While the N- and C-terminal domains are different, the core portion of the proteins consisting of the transmembrane and CBS-pair domains are highly similar in all organisms. Furthermore, genetic and physiological evidence in both prokaryotes and eukaryotes supports a role for the proteins in Mg²⁺ transport.

As requested, all mentions of MtCNNM in the text have been changed to MtCorB and CNNM has been removed from the title.

Nevertheless, my major concerns refer to concrete aspects of the structural interpretation that directly affect the transport mechanism and the transitions from the apo- to the MgATP-bound state as detailed below:

(1) Asymmetry of the MtCNNM dimer.

In the MgATP-bound MtCNNM crystals, the protein self-associate in non-symmetric dimers. In contrast, the nude MtCNNM dimers appear to maintain pseudo-2-fold symmetry. This point is key, as the cytoplasmic and intermembrane domains when analyzed pair-by-pair with their complementary counterparts in both types of conformers, appear to maintain local 2-fold symmetry. Previous studies by Gehring's team and other laboratories also indicate that CNNMs dimers likely form 2-fold symmetric assemblies. For example, crystals of the isolated CBS-domain pair (Bateman module) from CNNM2,

CNNM3, and CNNM4, no matter whether they were grown in the absence or in the presence of MgATP and/or PRL phosphatases, clearly form 2-fold symmetric assemblies. Pointing in the same direction, isolated CNBH domains from CNNM2, CNNM3, and CNNM4 associate in 2-fold dimers. Similarly, the DUF21 domains in MtCNNM are related by a 2-fold axis, both in the absence and in the presence of MgATP. A 180 degrees rotation also relates the Bateman modules in MgATP-bound MtCNNM. Moreover, solution studies on the cytoplasmic region of CNNM4 (Giménez-Mascarell, 2019) by SAXS recently showed 2-fold symmetric assemblies. Taken together, all these data invite us to think that the lack of symmetry observed in the MgATP-MtCNNM dimers might derive from the interactions with neighboring molecules in the crystals and facilitated by the flexibility and length of the linkers connecting the different protein domains (for example the AHB helices with the DUF21 domain, or the latter with the DUF21 domain). In fact, the model proposed in Figure 5e reflects a transition based on symmetric dimers. A similar disruption of the 2-fold symmetry, permitted by the flexibility and length of the linker connecting the CBS-pair domain with the CNBH domain, could also explain the asymmetry formerly observed in crystals of the cytoplasmic domain of CNNM2 (PDB entry 6N7E, Chen et al, Structure, 28, 324-335, 2020).

Thus, it is important to take into account that the asymmetry observed in the crystals could be artifactual and not maintained in solution, where no neighboring molecules can distort the dimer. A comparative MD analysis considering symmetric dimers (like for example the one depicted in the attached pictures) should be performed and discussed in the manuscript. For example, the authors should explore by MD whether the orientation of the AHB helix of molecule B containing residues (166-181) gets anomalously trapped by helix 35-46 from the DUF21 domain, thus deriving the dimer towards an asymmetric assembly. The transition from the apo- to the ATP-bound form should also be revisited and compared with the asymmetric option.

We agree that the asymmetry in the arrangement of the transmembrane and CBS-pair domains is a fascinating feature of the structures. As the reviewer correctly points out, this likely arises from crystal packing but also indicates the existence of flexibility and potentially dynamics of interface between the domains. We apologize for not properly addressing the potential consequences of crystal packing forces in the initial version of the paper. We have revised the text (bottom of page 7) to clearly state that the asymmetry likely arises from crystal packing.

We thank the reviewer for suggesting and providing the symmetric dimer. We have added an unbiased MD simulation with the symmetric dimer in the “Molecular dynamics simulations” section. In contrast to the simulation with the asymmetric dimer, we observed less flexibility and movements in the soluble domains throughout the 124 ns simulation. We also observed upward movement of AHB towards the TMD, suggesting entrapment and stabilization of the AHB by the helix-turn-helix motif of TMD. However, these *in silico* results are speculative, and future experiments using techniques without crystals, such as cryo-EM or SAXS (Giménez-Mascarell et al. 2019; PMID: 31842432), are needed to characterize the degree of asymmetry and conformational flexibility in CorB transporters.

(2) Effect of mutation R235L: Interestingly, mutation R235L impairs ATP binding at the CBS domains but does not appear to cause further effects in the Bateman module of MtCNNM. Under this scenario, how do the authors explain the dramatic reorientation of the Bateman modules and the drastic conformational change experienced by the entire dimer? It is well documented (for example visit articles describing the effect of nucleotides in IMPDH, CBS, or CNNMs) that binding of nucleotides at Bateman modules disrupts the interactions previously existing between the two CBS motifs. These changes are known to determine their relative orientation and their contact with the rest of the protein. This aspect should be discussed in the revised version of the manuscript.

We have added additional text in the discussion section commenting on the well-documented examples of dramatic dimeric rearrangement of CBS-pair domains upon nucleotide binding in other CBS domain-containing proteins as well as human CNNMs. Additionally, we have proposed an inactivation mechanism by the R235L mutant that may explain the lack of structural propagation towards the TMD.

(3) A movie showing the transformation from the unbound to the MgATP dimer should be provided as Supplemental Material. This would help readers understand the transitions experienced by the protein upon binding ATP.

We thank the reviewer for the suggestion. We have now provided a movie as Supplemental Movie 1, which illustrates the transformation from the unbound to the Mg²⁺-ATP dimer.

(4) Minor issue: The authors have placed a glycerol molecule in the model refined from the low-resolution C2 crystals. In fact, the superimposition of the two protein monomers suggests that the electron density has been wrongly attributed to a glycerol molecule, which coincides with part of the protein linker in the other monomer. Thus, this glycerol is very likely not present. Please correct this error in the PDB file, refine the structure accordingly and validate the model again.

We have collected a new dataset for the R235L structure and improved the resolution to 3.8Å, which allows more confident building of the AHB region. Indeed, where the glycerol molecule resided was actually a portion of the AHB linker. We thank the reviewer for pointing out the error, which has now been corrected.

REVIEWER COMMENTS

Reviewer #1 (Remarks to the Author):

congratulations to the authors for greatly improving this manuscript. a few additional comments and considerations:

- 1) too many acronyms in the abstract (AUC, HDX, MD, etc), please consider spelling out or phrasing a different way
- 2) too generous use of the phrase unambiguous; while I appreciate the results from your liposome assay, you in fact have not absolutely ruled out the possibility that a contaminating protein may be responsible for the measured activity (albeit unlikely).
- 3) you do not demonstrate any features of apparent selectivity or pharmacology, e.g. are Ca^{2+} , Co^{2+} , Ni^{2+} also transported? Does Cobalt-Hexammine block activity? These are clear remaining deficiencies in the present study and outstanding questions for the field - it would be nice if some of these could be addressed
- 4) ~25% identity is not "highly homologous to human" IMO; it's actually quite low
- 5) did you really assign the Mg^{2+} in your structure based only on the Huang et al report? Did the distances and geometry not match that expected for Mg^{2+} coordination? Did Mg^{2+} not refine as expected (compared to Na^{+} or Ca^{2+} for example). Please clarify.
- 6) it is not clear if 2 Mg^{2+} are bound simultaneously to each monomer within the dimer - it seems as if this is the case - and would be a remarkable feature. Can the authors comment on the distance between these 2 assigned Mg^{2+} ions? And do you in fact expect it is physically reasonable for them to be both bound and not repel each other? Do your maps or refined Bfactors suggest 50% occupancy? Please comment.
- 7) "The negative charge of the AHB is well conserved across evolution with the exception of yeast and roundworms" Any speculation as to why? Is this feature somehow redundant or not essential?
- 8) "S26A, N76A, E33A, P121G, and K122A equivalent mutants cannot be compared as the former two had localization problem and the later three were not tested" What do you mean by localization problem? Please clarify this sentence.

Reviewer #2 (Remarks to the Author):

First of all, I thank the authors for a comprehensive revision of their manuscript. With the performed comparison with the published results it is easier to judge whether any advancements are present with the current contribution, and certainly there are, although they might be more of interest for the specialised fields, but I leave it to the editor to judge the novelty.

I especially appreciate the in vitro assay, however in the contrast to the authors I don't feel like it answers the 'biggest controversy', but rather makes the whole thing even more complex, which requires additional experiments or at least a thorough discussion. Namely, if CorB proteins are indeed capable to transport Mg^{2+} on its own, what is the role of ATP-binding domains? If they do not provide the energy for the transport via ATP hydrolysis, are they involved in the transport regulation? if yes, how? Do they hydrolyse ATP at all (looking from Fig. 4 all necessary residues are in place). How is the hydrolysis coupled to the transport? Was the in vitro assay performed in the absence or in the presence of ATP? What if ATP is replaced with AMP-PNP? It seems like the most interesting part of the manuscript - at least for me, but I'm sure for many other people as well, so more discussion is needed.

Other less important concerns - the authors pinpoint that the work from Huang et al is based on the fragment (it is better to say on a truncated protein), however in their own case it is also truncated! I agree it is less truncated but it is not very fair to claim that you have it (almost) full-length. Clearly the nice finding is that in the present work the connection between TMD and AHB is visible, which unambiguously shows the connectivity between domains.

In the results section, the claim is made that MtCorB and TtCorB are highly homologous to human

CNNM2, but for me the sequence identity of 25% is pretty low, so better to say that key residues are conserved.

On page 7, the authors speculate that the asymmetric arrangement they observe can be a result of crystal packing, it would be good to show it (the packing) in the supplementary to see whether this is the case.

On page 8, where magnesium ion coordination is discussed, the interaction distances should be given as it can help to evaluate whether the ion is dehydrated or not.

Page 9, the observed UDM densities probably correspond to conserved positions of lipids, so the authors at least should mention that as lipids seem to play important roles in assisting conformational changes during transport.

Page 13. Liposome assays - is it indeed the assay developed by the authors or they adjusted the published assay to their target protein? if the latter, please cite the original publication (s).

Page 18, Discussion, please give the volume of the cavity for ions.

page 19. The authors claim that their 'results clearly demonstrate that CorB proteins are magnesium transporters themselves other than regulators of other transporters' but I don't see any evidence with their results excluding the latter possibility.

Fig. 2 - panel f, I112 seems to be also invariantly conserved

Table 1, from the values seems like the authors could try to extend the resolution even further.

Page 31, reconstitution procedure, 1 freeze-thaw cycle doesn't seem to be a good practice during prep

Reviewer #3 (Remarks to the Author):

In my opinion, the authors have addressed almost all requests satisfactorily, and the manuscript is now substantially improved. Accordingly, I recommend the publication of these interesting findings.

REVIEWER COMMENTS

Reviewer #1 (Remarks to the Author):

congratulations to the authors for greatly improving this manuscript. a few additional comments and considerations:

We thank the reviewer for the critical comments and the opportunity to improve our manuscript.

1) too many acronyms in the abstract (AUC, HDX, MD, etc), please consider spelling out or phrasing a different way

These acronyms have now been spelt out in the abstract.

2) too generous use of the phrase unambiguous; while I appreciate the results from your liposome assay, you in fact have not absolutely ruled out the possibility that a contaminating protein may be responsible for the measured activity (albeit unlikely).

The term “unambiguously” has now been removed when describing the liposome assay results.

3) you do not demonstrate any features of apparent selectivity or pharmacology, e.g. are Ca^{2+} , Co^{2+} , Ni^{2+} also transported? Does Cobalt-Hexammine block activity? These are clear remaining deficiencies in the present study and outstanding questions for the field - it would be nice if some of these could be addressed

We have added a panel to Supplementary Figure 8 showing the transport specificity of TtCorB for Mg^{2+} , Ca^{2+} , and Zn^{2+} . Using our assay to compare the transport properties of different ions is complicated by the widely varying affinity and fluorescence response of mag-fura-2 to the different ions (see <https://www.thermofisher.com/order/catalog/product/M1290>). To avoid these issues, we compare TtCorB transport with that of TmCorA reconstituted under the same conditions (1:30 lipid to protein ratio). The new panel e, reproduced below, shows that TtCorB and TmCorA have comparable transport activity for Mg^{2+} . However, TtCorB is more selective than TmCorA for Ca^{2+} and Zn^{2+} . Due to the weak fluorescence of mag-fura2 with bound Mn^{2+} , Co^{2+} and Ni^{2+} ions, we did not test them. Future studies using alternative dyes or techniques such as radioactive isotopes will allow us to confirm and extend these findings.

Supplemental Figure 8e, Comparison of ion specificity for TtCorB and TmCorA. The graph shows the relative change in fluorescence ratio of excitation at 330 nm over 369 nm in proteoliposomes or empty liposomes 10 min after addition of the indicated ions.

4) ~25% identity is not "highly homologous to human" IMO; it's actually quite low

We agree. The word "highly" has now been removed from this sentence.

5) did you really assign the Mg²⁺ in your structure based only on the Huang et al report? Did the distances and geometry not match that expected for Mg²⁺ coordination? Did Mg²⁺ not refine as expected (compared to Na⁺ or Ca²⁺ for example). Please clarify.

The MtCorBAC + Mg²⁺-ATP structure was solved at 3.25 Å, which is at the borderline for visualizing ions such as Mg²⁺ or Na⁺. In fact, we only observed density in one of the transmembrane domains (chain A). Both Mg²⁺ and Na⁺ are candidates for this density, since they are present in our crystallization condition (100 mM sodium citrate, 100 mM NaCl, and 20 mM MgCl₂). We have tried refining the structure with either Mg²⁺ or Na⁺, and we did not observe noticeable difference in B-factors or coordination. This is probably because Mg²⁺ and Na⁺ have similar coordination geometry and distances. The coordination distances range from 1.9 to 2.7 Å, due to the limited resolution (3.25 Å) and lack of definition of the side chains. Since the distances span the typical values for Mg²⁺ (2.1 Å) and Na⁺ (2.4 Å), we left them unrestrained. The decision to label the density in the initial PDB deposition as Na⁺ was based on the analysis by the Check My Metal Server (<https://cmm.minorlab.org/>), which showed somewhat better fit with Na⁺.

We did attempt to resolve the issue using anomalous scattering and collected anomalous datasets of MtCorBAC crystals co-crystallized with Co²⁺ or Mn²⁺. We did observe anomalous scattering at the Mg²⁺-ATP binding sites in the CBS-pair domain but did not see a signal at the transmembrane domain site. This suggests that the site is either highly selective for Mg²⁺ or it does not bind divalent cations. Similar results were obtained with Rb⁺ or Cs⁺ soaked crystals. Thus, for the time being, we cannot assign the density to either Mg²⁺ or Na⁺. As Haung et al. identified a Mg²⁺ ion at that site in their 2.0 Å structure of the bacterial ortholog, we labeled the density as Mg²⁺ in our structure.

6) it is not clear if 2 Mg²⁺ are bound simultaneously to each monomer within the dimer - it seems as if this is the case - and would be a remarkable feature. Can the authors comment on the distance between these 2 assigned Mg²⁺ ions? And do you in fact expect it is physically reasonable for them to be both bound and not repel each other? Do your maps or refined Bfactors suggest 50% occupancy? Please comment.

The reviewer is correct. Since the TMD is a dimer, we would expect to see two bound ions but in fact, we only observed Mg²⁺ bound to one polypeptide chain. The reason for this is not clear. It could be result from a subtle conformational difference between the two monomers. A more conservative interpretation is that we lacked sufficient resolution to clearly see both ions. In any case, if both Mg²⁺ are bound simultaneously, the distance between them would be 13 Å. Since they are extensively coordinated by negatively charged residues, we would not expect them to repel each other. The structure was modeled with 100% occupancy for the Mg²⁺ ion.

7) "The negative charge of the AHB is well conserved across evolution with the exception of yeast and roundworms" Any speculation as to why? Is this feature somehow redundant or not essential?

The AHB domains from plants and archaea are particularly rich in negatively charged amino acids. It is unclear why other species have fewer acidic residues, but we expect that some positions, such as those conserved between humans and archaea (Fig. 7a), are more important than others. As some organisms have multiple CNNM homologs (*C. elegans* has five), it is also

possible that the CNNM proteins have been tuned to different levels of activity or response to regulatory signals.

8) "S26A, N76A, E33A, P121G, and K122A equivalent mutants cannot be compared as the former two had localization problem and the later three were not tested" What do you mean by localization problem? Please clarify this sentence.

What we meant to say is that these five mutations were not tested by Huang et al. The S26A and N76A equivalent mutants did not localize to the cell membrane, while E33A, P121G, and K122A were not tested by them. We have now clarified this statement to simply state that the Huang et al. did not assay transport in the five mutations.

Reviewer #2 (Remarks to the Author):

First of all, I thank the authors for a comprehensive revision of their manuscript. With the performed comparison with the published results it is easier to judge whether any advancements are present with the current contribution, and certainly there are, although they might be more of interest for the specialised fields, but I leave it to the editor to judge the novelty.

We thank the reviewer for acknowledging our efforts and providing critical comments for improving our manuscript.

I especially appreciate the *in vitro* assay, however in the contrast to the authors I don't feel like it answers the 'biggest controversy', but rather makes the whole thing even more complex, which requires additional experiments or at least a thorough discussion. Namely, if CorB proteins are indeed capable to transport Mg²⁺ on its own, what is the role of ATP-binding domains? If they do not provide the energy for the transport via ATP hydrolysis, are they involved in the transport regulation? if yes, how? Do they hydrolyse ATP at all (looking from Fig. 4 all necessary residues are in place). How is the hydrolysis coupled to the transport? Was the *in vitro* assay performed in the absence or in the presence of ATP? What if ATP is replaced with AMP-PNP? It seems like the most interesting part of the manuscript - at least for me, but I'm sure for many other people as well, so more discussion is needed.

We thank the reviewer for the positive reception to our *in vitro* assay. Indeed, we also wonder about the role of ATP binding in CNNM/CorB function. We would like to point out that ATP was not added in our *in vitro* assay. This suggests that either the protein purified from *E. coli* still retains the ATP after purification or it is not required for activity in the liposome assay.

As for whether CorB proteins hydrolyze ATP, we believe they do not. CBS-pair domain exists in many proteins, such as AMPK, CIC channels, MgtE magnesium channel, but there are no reports that the CBS-pair domain possesses ATPase activity. ATP binding to the CBS-pair domain of MgtE is thought to play a regulatory role (Tomita et al. Nat Commun 8:148, 2017).

To directly address the question, we measured ATPase activity by full-length MtCorB and TtCorB using the same assay and conditions as in our previous paper (Chen et al., Structure 28:324, 2020). We did not observe any appreciable level of ATPase activity (sp. act. of < 1 h⁻¹) (Response Figure 1), in agreement with the previous studies with eukaryotic and bacterial CNNM/CorB proteins. Huang et al. (2021) failed to detect ATPase assay with the bacterial protein, TpCorB and we observed no ATPase activity with the cytosolic fragments of human CNNM2-4 (Chen et al., Structure 28:324, 2020). We have expanded the discussion of ATP binding on page 19 of the manuscript.

Response Figure 1. ATP hydrolysis assay of CorB proteins. ATP hydrolysis assay was performed using Malachite Green Phosphatase Assay Kit. The assays were performed in a 100 μ L final reaction volume consisting of 1 μ M CorB proteins or 0.03 μ M (0.1 unit) of apyrase protein as positive control in presence of 20 mM $MgCl_2$ and 1 mM ATP. The reaction was performed at RT for 30 min. The measurements were performed in triplicate.

Other less important concerns - the authors pinpoint that the work from Huang et al is based on the fragment (it is better to say on a truncated protein), however in their own case it is also truncated! I agree it is less truncated but it is not very fair to claim that you have it (almost) full-length. Clearly the nice finding is that in the present work the connection between TMD and AHB is visible, which unambiguously shows the connectivity between domains.

The term “near complete” has now been removed.

In the results section, the claim is made that MtCorB and TtCorB are highly homologous to human CNNM2, but for me the sequence identity of 25% is pretty low, so better to say that key residues are conserved.

The word “highly” has now been removed from this sentence.

On page 7, the authors speculate that the asymmetric arrangement they observe can be a result of crystal packing, it would be good to show it (the packing) in the supplementary to see whether this is the case.

A figure showing the crystal packing has now been added as Supplementary Figure 3d.

On page 8, where magnesium ion coordination is discussed, the interaction distances should be given as it can help to evaluate whether the ion is dehydrated or not.

The interaction distances have now been added to Figure 2d.

Page 9, the observed UDM densities probably correspond to conserved positions of lipids, so the authors at least should mention that as lipids seem to play important roles in assisting conformational changes during transport.

We thank the reviewer for the suggestion. We have now added a statement mentioning that these detergent molecules likely correspond to conserved lipid binding sites important for assisting conformational changes during transport.

Page 13. Liposome assays - is it indeed the assay developed by the authors or they adjusted the published assay to their target protein? if the latter, please cite the original publication (s).

Our liposome assays were designed by modifying published assays previously done for CorA proteins (Payandeh et al., J Biol Chem 283:11721, 2008) & (Stetsenko et al., Sci Rep 10:840, 2020). We have now included these two citations in the beginning of this section.

Page 18, Discussion, please give the volume of the cavity for ions.

The volume of the cavity ($\sim 1,500 \text{ \AA}^3$) has now been indicated.

page 19. The authors claim that their 'results clearly demonstrate that CorB proteins are magnesium transporters themselves other than regulators of other transporters' but I don't see any evidence with their results excluding the latter possibility.

This sentence has been rephrased to remove the latter statement.

Fig. 2 - panel f, I112 seems to be also invariantly conserved

Yes, we thank the reviewer for pointing this out. This has now been added to the text.

Table 1, from the values seems like the authors could try to extend the resolution even further.

We thank the reviewer for the suggestion and have extended the resolution for the MtCorB_{CBS} dataset further to 2.05 Å.

Page 31, reconstitution procedure, 1 freeze-thaw cycle doesn't seem to be a good practice during prep

During optimization of the liposome assay, we tried varying number of freeze-thaw cycles, and we found that increasing number of freeze-thaw cycles reduces the activity of MtCorB. The exact reason for this is unknown, but we suspect that the increased number of freeze-thaw cycles might increase the proportion of inactive transporters through destabilizing or affecting their orientation in the liposome. Therefore, in the end, we decided to maximize our activity by doing one freeze-thaw cycle. This has now been noted in the Methods section.

Reviewer #3 (Remarks to the Author):

In my opinion, the authors have addressed almost all requests satisfactorily, and the manuscript is now substantially improved. Accordingly, I recommend the publication of these interesting findings.

We thank the reviewer for the recommendation.

REVIEWERS' COMMENTS

Reviewer #2 (Remarks to the Author):

I think that the authors did an excellent job and the latest version is suitable for publication in Nature Communications.

REVIEWERS' COMMENTS

Reviewer #2 (Remarks to the Author):

I think that the authors did an excellent job and the latest version is suitable for publication in Nature Communications.

We thank the reviewer for the recommendation.